# Conditional Diffusion Anomaly Modeling on Graphs

**Chunyu Wei[1], Haozhe Lin[2], Yueguo Chen[1]\*, Yunhai Wang[1]**
[1]Renmin University of China, China
[2]Tsinghua University, China
`weicy15@icloud.com`
`linhz@mail.tsinghua.edu.cn`
`chenyueguo@ruc.edu.cn`
`cloudseawang@gmail.com`

## Abstract

Graph anomaly detection (GAD) has become a critical research area, with successful applications in financial fraud and telecommunications. Traditional Graph Neural Networks (GNNs) face significant challenges: at the topology level, they suffer from over-smoothing that averages out anomalous signals; at the feature level, discriminative models struggle when fraudulent nodes obfuscate their features to evade detection. In this paper, we propose a Conditional Graph Anomaly Diffusion Model (CGADM) that addresses these issues through the iterative refinement and denoising reconstruction properties of diffusion models. Our approach incorporates a prior-guided diffusion process that injects a pre-trained conditional anomaly estimator into both forward and reverse diffusion chains, enabling more accurate anomaly detection. For computational efficiency on large-scale graphs, we introduce a prior confidence-aware mechanism that adaptively determines the number of reverse denoising steps based on prior confidence. Experimental results on benchmark datasets demonstrate that CGADM achieves state-of-the-art performance while maintaining significant computational advantages for large-scale graph applications. [2]

## 1 Introduction

Graph anomaly detection (GAD) has become a critical research area, with successful applications in financial fraud detection [Huang et al., 2022, Dou et al., 2020] and telecommunication fraud detection [Yang et al., 2021]. Graph Neural Networks (GNNs) have gained prominence for GAD due to their ability to model topological structures through message passing, which aggregates neighborhood information to generate node representations that are then classified as normal or anomalous [Kipf and Welling, 2017, Hamilton et al., 2017, Velickovic et al., 2018, Xu et al., 2019].

However, discriminative models based on feature aggregation exhibit inherent shortcomings.

1. From **topology**-level perspective, vanilla GNNs suffer from over-smoothing, acting as low-pass filters that average anomalous representations, making them less distinguishable. As illustrated in the left part of Figure 1, fraudulent nodes exploit this by strategically connecting with carefully selected neighbors to disguise their anomalous patterns. For instance, in money laundering transactions, fraudsters can distribute transactions or create numerous interactions with bot accounts to blend in with the crowd.

---

\*Corresponding author. He works at Big Data and Responsible Artificial Intelligence for National Governance, Renmin University of China
[2]The code is available on https://github.com/weicy15/CGADM.

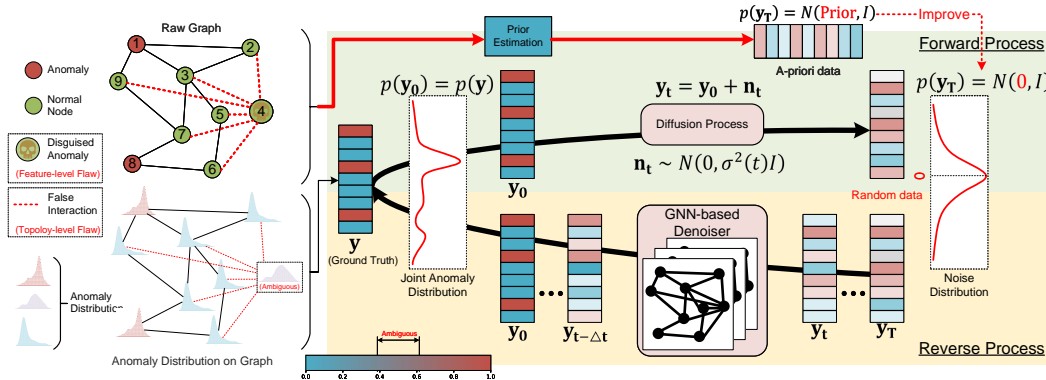

Figure 1: An illustration of Generative Graph Anomaly Detection.

2. From **feature**-level perspective, discriminative models detect anomalies by learning decision boundaries between normal and anomalous points. As fraudulent nodes evolve and obfuscate their features, they can cross these boundaries, evading detection.

Diffusion models (DMs) can address these limitations through their two key properties: **iterative refinement** and **denoising reconstruction**. Iterative refinement applies GNN-based denoisers that incorporate neighborhood information while preserving high-frequency anomaly signals via residual propagation, preventing over-smoothing. Meanwhile, denoising reconstruction recovers underlying anomaly patterns even when nodes disguise their features. (See Appendix R for theoretical analysis).

Applying DMs for GAD introduces two major challenges, as shown in right part of Figure 1:

**Effectiveness.** Traditional denoising models have primarily focused on unconditional generative modeling [Song and Ermon, 2019, Song et al., 2021b, Ramesh et al., 2022]. While many tasks in the image or video domain have introduced guided-diffusion models to generate photo-realistic images that match the semantic meanings or content of the label, text, or corrupted images, most work in the graph domain has started generating from white noise or empty or fully connected graphs. However, for anomaly detection on graphs, due to various deceptive and obfuscating tactics employed by anomalous nodes, directly recovering the underlying true distribution from a random noise distribution may not yield satisfactory results.

**Efficiency.** The reverse process of DMs requires numerous iterative denoising samplings [Yi et al., 2023, Chen et al., 2023b]. Existing graph diffusion models utilize a GNN-based encoder to update all nodes at time step $t$ during each iterative refinement to obtain the nodes at time step $t - 1$. While this approach is feasible for standard graph generation tasks, it becomes computationally prohibitive for anomaly detection tasks on extremely large graphs. Performing such iterative operations across potentially millions of nodes in the entire graph can significantly increase computational overhead, thereby affecting the practical applicability of the algorithm.

We propose a novel Conditional Graph Anomaly Diffusion Model (CGADM) for graph anomaly detection to address the aforementioned challenges synergistically. Unlike existing diffusion-based approaches that performing data augmentation to address class imbalance, CGADM directly generates anomaly judgments through joint distribution modeling, representing a fundamentally new model-centric paradigm for GAD.

To tackle the effectiveness issue, we propose a prior-guided diffusion process, which injects a pre-trained conditional anomaly estimator into both the forward and reverse diffusion chains. This approach constructs a denoising diffusion probabilistic model for more accurate anomaly detection. Specifically, we introduce a lightweight model to estimate an anomaly prior for each node, serving as the endpoint for our forward noise addition process and the starting point for our reverse denoising process. Based on this new probabilistic model, we redesign the probability model and optimization objective of our CGADM.

To tackle the efficiency issue, we build on the intuition that normal nodes are generally farther from the decision boundary compared to anomalous nodes that have narrowly evaded detection. Therefore,

in the reverse process, we introduce a prior confidence-aware mechanism to adaptively determine the reverse time step for each node. Nodes with high confidence in their anomaly prior require fewer time steps, while those with lower confidence require more sampling time steps. This approach not only accurately estimates the anomaly probability for each node but also reduces the number of predictions in the reverse process, thereby decreasing computational time.

Through experiments on benchmarks for GAD, CGADM achieves state-of-the-art results. Additional studies confirm the computational advantages of our framework.

## 2 Related Work

### 2.1 Graph Anomaly Detection

Graph anomaly detection [Duan et al., 2023] aims to identify nodes that deviate significantly from most other nodes. Various GNN-based methods have been proposed to address this challenge. Early approaches like FdGars [Wang et al., 2019b] and CARE-GNN [Dou et al., 2020] focused on user classification and neighbor aggregation respectively. Follow-up works tackled specific issues: FRAUDRE [Zhang et al., 2021] and PC-GNN [Liu et al., 2021b] addressed class imbalance, while AMNet [Chai et al., 2022], BWGNN [Tang et al., 2022], and GHRN [Gao et al., 2023b] improved feature handling through frequency-based approaches.

Recent advancements have explored novel directions: GDN [Gao et al., 2023a] addressed structural distribution shifts, SEC-GFD [Xu et al., 2024] handled heterophily via spectral filtering, GGAD [Qiao et al., 2024] generated pseudo-anomalies, and ADA-GAD [He et al., 2024] mitigated anomaly overfitting. Unlike these approaches, our CGADM introduces a generative diffusion framework that models the joint anomaly distribution over the graph, enabling holistic detection without relying on augmentation strategies Wei et al. [2023b].

However, existing methods Wei et al. [2025, 2023c] rely on discriminative models with feature aggregation, making them vulnerable to over-smoothing Wei et al. [2022d] and camouflage tactics. Our approach departs from this paradigm by proposing a generative model that jointly models the anomaly distribution of each node on the graph.

### 2.2 Diffusion Model

Denoising diffusion probabilistic models (DDPMs) [Ho et al., 2020, Song et al., 2021a], or simply diffusion models, are a class of probabilistic generative models that transform noise into data samples, hence primarily used for generative tasks [Dhariwal and Nichol, 2021, Rombach et al., 2022]. Diffusion-based generative models have demonstrated strong capabilities in generating high-quality graphs [Niu et al., 2020, Liu et al., 2019, Jo et al., 2022, Haefeli et al., 2022, Chen et al., 2022, Vignac et al., 2023, Kong et al., 2023]. Haefeli et al. [2022] designed a model limited to graphs without attributes and similarly observed the benefits of discrete diffusion for graph generation. Previous graph diffusion models were based on Gaussian noise. Niu et al. [2020] generated adjacency matrices indicating the presence of edges by thresholding continuous values, while Jo et al. [2022] extended this model to handle node and edge attributes. Digress [Vignac et al., 2023] was the first to propose a discrete diffusion model for graphs. Regarding the severe label imbalance problem in anomaly detection, many existing anomaly detection methods improve datasets by generating synthetic anomalies [Chen et al., 2020b, Ding et al., 2020], creating a more balanced environment.

We approaches from a different angle, using diffusion models to model the joint distribution of anomalies on large-scale graphs for more precise and robust anomaly detection.

## 3 Preliminaries

**Attributed Graph.** An attributed graph is denoted as $\mathcal{G} = \{\mathcal{V}, \mathcal{E}, \mathbf{X}\}$, where $\mathcal{V} = \{v_1, v_2, \ldots, v_N\}$ represents the set of all $N$ nodes on graph $\mathcal{G}$, and $\mathcal{E} = \{e_{ij}|v_i, v_j \in \mathcal{V}\}$ signifies the set of edges, indicating the existence of an edge between nodes $v_i$ and $v_j$. For each node $v_i$, there exists a $d$-dimensional feature vector, $x_i \in \mathbb{R}^d$. The feature vectors of all nodes form the feature matrix of the graph, denoted as $\mathbf{X} = [x_1, x_2, \ldots, x_N] \in \mathbb{R}^{N \times d}$. An adjacency matrix $\mathbf{A}$ records the relationships between nodes on graph $\mathcal{G}$. Each entry $\mathbf{A}_{ij} = 1$ if there exists $e_{ij} \in \mathcal{E}$, otherwise, $\mathbf{A}_{ij} = 0$.

**Anomaly Detection on Graph.** Consider two disjoint subsets of $\mathcal{V}$, namely $\mathcal{V}_a$ and $\mathcal{V}_n$, such that $\mathcal{V}_a \cap \mathcal{V}_n = \emptyset$. $\mathcal{V}_a$ contains all nodes labeled as anomalous, and $\mathcal{V}_n$ comprises all normal nodes. The goal of graph anomaly detection (GAD) is to compute anomaly probability $p(\mathbf{y}|\mathcal{E}, \mathbf{X})$ of the unlabeled nodes with partial node labels. Please refer Appendix F for challenges of GAD.

**Diffusion Probabilistic Model.** An efficient diffusion model must satisfy three key properties: (1) The conditional distribution $q(z_t|x)$ should possess a closed-form equation to circumvent the recursive application of noise during training. (2) The posterior $q(z_{t-1}|z_t, x)$ should also have a closed-form solution to serve as the neural network's target. (3) The limiting distribution $q_\infty = \lim_{T\to\infty} q(z_T|x)$ should be independent of $x$, enabling its use as a prior distribution for inference. These properties are all met when the noise follows a Gaussian distribution. The common steps in the diffusion model are shown in Appendix B.

# 4 Methodology

We formulate the GAD problem as a task of modeling the joint conditional distribution of anomalies on the graph. This prior distribution serves as the endpoint for adding noise and the starting point for inference. CGADM gradually transforms the ground truth anomaly distribution into the prior distribution instead of the conventional Guassian distribution. By utilizing a topological-guided denoising network, CGADM is capable of simultaneously modeling the topological information and features of nodes to iteratively recover the ground truth. To expedite the inference process, we introduce a prior-aware strided sampling strategy. To enable inference over arbitrary numbers of steps, we propose a conditional non-Markovian reverse process.

## 4.1 Diffuse Ground Truth to Prior

In light of Section 3, we propose to cast the graph anomaly detection problem as a generative task. We set $\mathbf{y}_0$ as the anomaly ground truth and $\mathbf{y}_{1:T}$ as the intermediate predictions generated in the forward process of the diffusion model. The objective of graph anomaly detection then becomes the maximization of the log-likelihood $p(\mathbf{y}_0|\mathcal{E}, \mathbf{X})$. Consequently, Equation 2 can be restructured as the following Conditional Evidence Lower Bound (CELBO) to serve as our new optimization target:

$$\log p_\theta(\mathbf{y}_0|\mathcal{E}, \mathbf{X}) = \log \int p_\theta(\mathbf{y}_{0:T}|\mathcal{E}, \mathbf{X})d\mathbf{y}_{1:T} \geq \mathbb{E}_{q(\mathbf{y}_{1:T}|\mathbf{y}_0, \mathcal{E}, \mathbf{X})}\left[\log \frac{p_\theta(\mathbf{y}_{0:T}|\mathcal{E}, \mathbf{X})}{q(\mathbf{y}_{1:T}|\mathbf{y}_0, \mathcal{E}, \mathbf{X})}\right], \quad (1)$$

where $p_\theta(\mathbf{y}_{0:T}|\mathcal{E}, \mathbf{X})$ is the joint distribution of the target and the predictions under the denoising model parameters $\theta$, and $q(\mathbf{y}_{1:T}|\mathbf{y}_0, \mathcal{E}, \mathbf{X})$ is the conditional distribution of forward process given the ground truth and the input data.

By substituting Equation 1 into Equation 17, we can express our optimization objective as follows:

$$\mathcal{L} = \mathbb{E}_q\left[-\log p_\theta(\mathbf{y}_0|\mathbf{y}_1, \mathcal{E}, \mathbf{X})\right] + \mathbb{E}_q\left[\mathbb{D}_{KL}\left(q(\mathbf{y}_T|\mathbf{y}_0, \mathcal{E}, \mathbf{X}) \| p(\mathbf{y}_T|\mathcal{E}, \mathbf{X})\right]\right.$$
$$+ \sum_{t=2}^{T} \mathbb{E}_q\left[\mathbb{D}_{KL}\left(q(\mathbf{y}_{t-1}|\mathbf{y}_t, \mathbf{y}_0, \mathcal{E}, \mathbf{X}) \| p_\theta(\mathbf{y}_{t-1}|\mathbf{y}_t, \mathcal{E}, \mathbf{X})\right)\right]. \quad (2)$$

Following the conventions of Denoising Diffusion Probabilistic Models (DDPM) [Ho et al., 2020], we respectively name the first, second, and third terms of the above objective function as the reconstruction term $\mathcal{L}_{recon}$, the prior matching term $\mathcal{L}_{prior}$, and the consistency term $\mathcal{L}_{con}$.

To avoid our CGADM recovering the joint anomaly distribution starting from random noise [Han et al., 2022b], we modify the endpoint of the diffusion process from the conventional Guassian distribution $N(0, I)$ to:

$$p(\mathbf{y}_T|\mathcal{E}, \mathbf{X}) = N(g_\phi(\mathcal{E}, \mathbf{X}), I), \quad (3)$$

where $g_\phi(\mathcal{E}, \mathbf{X})$ is a parameterized network pretrained on training set $D$ to estimate the mean value of the final normal distribution. By doing so, we effectively utilize the condition $\mathcal{E}, \mathbf{X}$ in the distribution $p(\mathbf{y}_T|\mathcal{E}, \mathbf{X})$ to help us establish a prior understanding of the joint anomaly distribution.

The prior matching term $\mathcal{L}_{prior}$ is a parameter-free term. In order to make it close to zero, we need to adjust the forward process in combination with the calculation of the prior $g_\phi(\mathcal{E}, \mathbf{X})$. Following the practice of Pandey et al. [2022], we define the noise-adding process at each step as follows:

$$q(\mathbf{y}_t|\mathbf{y}_{t-1}, g_\phi(\mathcal{E}, \mathbf{X})) = \mathcal{N}(\mathbf{y}_t; \sqrt{1-\beta_t}\mathbf{y}_{t-1} + (1-\sqrt{1-\beta_t})g_\phi(\mathcal{E}, \mathbf{X}), \beta_t I), \quad (4)$$

where $\mathcal{N}$ represents the Gaussian Distribution, and $\beta_t \in (0, 1)$ regulates the noise scales added at step $t$. This noise-adding step allows for a closed-form sampling distribution at any arbitrary timestep $t$, according to the additivity of the Gaussian distribution:

$$q(\mathbf{y}_t|\mathbf{y}_0, \mathcal{E}, \mathbf{X}) = q(\mathbf{y}_t|\mathbf{y}_0, g_\phi(\mathcal{E}, \mathbf{X})) = \mathcal{N}(\mathbf{y}_t; \sqrt{\bar{\alpha}_t}\mathbf{y}_0 + (1 - \sqrt{\bar{\alpha}_t})g_\phi(\mathcal{E}, \mathbf{X}), (1 - \bar{\alpha}_t)I), \quad (5)$$

where $\alpha_t := 1 - \beta_t$ and $\bar{\alpha}_t := \prod_t \alpha_t$. This sampling distribution enables $\mathcal{L}_{prior}$ to be close to zero when $t = T$. Intuitively, the noise-adding process defined by Equation 5 can be interpreted as an interpolation between the true data $\mathbf{y}_0$ and the estimated prior $g_\phi(\mathcal{E}, \mathbf{X})$, which exhibits a gradual transition from the true data towards the estimated prior over the course of the forward process.

With the above formulation, we can derive a tractable posterior that serves as the target for our denoising network. It can be expressed as follows:

$$q(\mathbf{y}_{t-1}|\mathbf{y}_t, \mathbf{y}_0, \mathcal{E}, \mathbf{X}) = q(\mathbf{y}_{t-1}|\mathbf{y}_t, \mathbf{y}_0, g_\phi(\mathcal{E}, \mathbf{X})) = \mathcal{N}\left(\mathbf{y}_{t-1}; \tilde{\mu}(\mathbf{y}_t, \mathbf{y}_0, g_\phi(\mathcal{E}, \mathbf{X})), \tilde{\beta}_t \mathbf{I}\right), \quad (6)$$

where $\tilde{\mu} := \gamma_0 \mathbf{y}_0 + \gamma_1 \mathbf{y}_t + \gamma_2 g_\phi(\mathcal{E}, \mathbf{X})$ and $\tilde{\beta}_t := \frac{1 - \bar{\alpha}_{t-1}}{1 - \bar{\alpha}_t}\beta_t$, with:

$$\gamma_0 = \sqrt{\beta_t \bar{\alpha}_{t-1}}, \quad \gamma_1 = \frac{(1 - \bar{\alpha}_{t-1})\sqrt{\alpha_t}}{(\alpha_t - 1)(\sqrt{\alpha_t} + \sqrt{\bar{\alpha}_{t-1}})}, \quad \gamma_2 = \frac{1}{1 - \bar{\alpha}_t}. \quad (7)$$

For detailed derivation, please refer to Appendix C.

## 4.2 Topological-guided Denoising Network

According to Equation 4, we define $p_\theta(\mathbf{y}_{t-1}|\mathbf{y}_t, \mathcal{E}, \mathbf{X})$ as $N(\mathbf{y}_{t-1}; \mu_\theta(\mathbf{y}_t, t, \mathcal{E}, \mathbf{X}), \Sigma_\theta(\mathbf{y}_t, t, \mathcal{E}, \mathbf{X}))$ for $1 < t \le T$. Following the setup of DDPM, we set $\Sigma_\theta(\mathbf{y}_t, t, \mathcal{E}, \mathbf{X}) = \sigma_t^2 \mathbf{I}$ to untrained time-dependent constants and set $\sigma_t^2 = \tilde{\beta}_t$. For the parameterization, we may select:

$$\mu_\theta(\mathbf{y}_t, t, \mathcal{E}, \mathbf{X}) = \frac{1}{\sqrt{\alpha_t}}(\mathbf{y}_t - \frac{\beta_t}{\sqrt{1 - \bar{\alpha}_t}}\epsilon_\theta(\mathbf{y}_t, t, \mathcal{E}, \mathbf{X})), \quad (8)$$

where $\epsilon_\theta$ is a parameterized network predicting the forward diffusion noise $\epsilon$ sampled for anomaly scores $\mathbf{y}_t$.

An anomalous node is typically strongly correlated not only with its node features but also with the its local topological structure. The bias brought about by a few anomalous nodes is high-frequency information in the frequency domain. Most existing GNNs act as low-pass filters and cannot effectively capture the high-frequency signals carried by anomalous nodes. Borrowing the idea from GCNII [Chen et al., 2020a], we adopt a residual propagation mechanism that prevents the high-frequency information of nodes from being overlooked due to over-smoothing in the multi-layer graph convolution process:

$$\mathbf{h}_v^l = \sigma\left(\mathbf{W}^{l-1}\left(\mathbf{h}_v^{l-1} - \frac{1}{|\mathcal{N}(v)|}\sum_{u \in \mathcal{N}(v)} \mathbf{h}_u^{l-1}\right)\right), \quad \mathbf{h}^{final} = AGG(\mathbf{h}_v^0, \mathbf{h}_v^1, \ldots, \mathbf{h}_v^L), \quad (9)$$

where $L$ is the number of graph convolution layers and $AGG(\cdot)$ can be a simple aggregation function such as summation or concatenation. With this message-passing mechanism, we define our topological-aware denoising network as $\epsilon_\theta(\mathbf{y}_t, t, \mathcal{E}, \mathbf{X}) = \epsilon_\theta(\mathbf{y}_t, t, \mathbf{H}^{final})$. For more details about the denoising network, please refer to Appendix H.

To execute our training, we sample $\mathbf{y}_t$ according to Equation 5. Through the reparameterization trick, we can derive:

$$\mathbf{y}_t = \sqrt{\bar{\alpha}_t}\mathbf{y}_0 + (1 - \sqrt{\bar{\alpha}_t})g_\phi(\mathcal{E}, \mathbf{X}) + \sqrt{1 - \bar{\alpha}_t}\epsilon. \quad (10)$$

We simplify $\mathcal{L}_{recon}$ and $\mathcal{L}_{con}$ to obtain the final loss $\mathcal{L}$:

$$\mathcal{L}_\epsilon = ||\epsilon - \epsilon_\theta(\sqrt{\bar{\alpha}_t}\mathbf{y}_0 + (1 - \sqrt{\bar{\alpha}_t})g_\phi(\mathcal{E}, \mathbf{X}) + \sqrt{1 - \bar{\alpha}_t}\epsilon, t, \mathcal{E}, \mathbf{X})||^2 \quad (11)$$

Where elements in $\mathbf{t}$ is uniformly distributed between 1 and $T$. The case of $t = 1$ corresponds to $\mathcal{L}_{recon}$. Similar to DDPM, the cases where $t > 1$ correspond to an unweighted version of $\mathcal{L}_{con}$. The whole process of training is shown in Appendix I.

## 4.3 Inference for Anomaly Detection

For image synthesis, DMs typically draw random Gaussian noise for the reverse process, with generation guided by pre-trained classifiers or other signals. However, for graph anomaly detection, generating directly from pure noise may not yield accurate results due to the deceptive tactics employed by anomalous nodes.

We propose an inference strategy that aligns with CGADM training, starting from a prior-guided initialization $\mathbf{y}_T \sim \mathcal{N}(g_\phi(\mathcal{E}, \mathbf{X}), I)$ rather than standard Gaussian noise. At each step $t$, we first estimate the denoised anomaly score:

$$\hat{\mathbf{y}}_0 = \frac{1}{\sqrt{\bar{\alpha}_t}}(\mathbf{y}_t - (1 - \sqrt{\bar{\alpha}_t})g_\phi(\mathcal{E}, \mathbf{X}) - \sqrt{1 - \bar{\alpha}_t}\epsilon_\theta(\mathbf{y}_t, t, \mathcal{E}, \mathbf{X})) \tag{12}$$

Then we use this estimate to predict the intermediate state: $\mathbf{y}_{t-1} = \gamma_0\hat{\mathbf{y}}_0 + \gamma_1\mathbf{y}_t + \gamma_2 g_\phi(\mathcal{E}, \mathbf{X}) + \tilde{\beta}_t z$, where $z \sim \mathcal{N}(0, I)$ and the coefficients $\gamma_0, \gamma_1, \gamma_2$ and $\tilde{\beta}_t$ are defined in Equation 6. This process iteratively refines the anomaly representations until we obtain the final anomaly scores $\mathbf{y}_0$. The complete algorithm is provided in Algorithm 1.

---

**Algorithm 1** Inference for Anomaly Detection

---

1: Initialize $\mathbf{y}_T \sim \mathcal{N}(g_\phi(\mathcal{E}, \mathbf{X}), I)$
2: **for** $t = T$ to 1 **do**
3:     Calculate reparameterized $\hat{\mathbf{y}}_0$ according to Equation 10:

$$\hat{\mathbf{y}}_0 = \frac{1}{\sqrt{\bar{\alpha}_t}}(\mathbf{y}_t - (1 - \sqrt{\bar{\alpha}_t})g_\phi(\mathcal{E}, \mathbf{X}) - \sqrt{1 - \bar{\alpha}_t}\epsilon_\theta(\mathbf{y}_t, t, \mathcal{E}, \mathbf{X})) \tag{13}$$

4:     **if** $t > 1$ **then**
5:         Draw $z \sim \mathcal{N}(0, I)$
6:         $\mathbf{y}_{t-1} = \gamma_0\hat{\mathbf{y}}_0 + \gamma_1\mathbf{y}_t + \gamma_2 g_\phi(\mathcal{E}, \mathbf{X}) + \tilde{\beta}_t z$, according to Equation 6.
7:     **else**
8:         Set $\mathbf{y}_{t-1} = \hat{\mathbf{y}}_0$
9:     **end if**
10: **end for**
11: **return** $y_0$

---

The key advantage of this approach is that it leverages our prior knowledge of anomaly patterns to guide the generation process, making it more resistant to deceptive tactics employed by anomalous nodes compared to generating directly from random noise.

## 4.4 Prior-aware Strided Sampling

As can be seen from Equation 11, our training actually results in a topological-aware denoising network capable of denoising the predicted prior score at arbitrary time step $t$. Inspired by Song et al. [2021a], we can use this denoising network to perform time-step skipping sampling, greatly reducing the number of sampling steps. By discarding the Markov constraint brought by Equation 4, we can obtain the conditional non-Markovian reverse process different from Equation 6 as follows:

$$\mathbf{y}_{t-1} = \sqrt{\bar{\alpha}_{t-1}}\hat{\mathbf{y}}_0 + (1 - \sqrt{\bar{\alpha}_{t-1}})g_\phi(\mathcal{E}, \mathbf{X}) + \sqrt{1 - \bar{\alpha}_{t-1} - \sigma_t^2}\epsilon_\theta(\mathbf{y}_t, t, \mathcal{E}, \mathbf{X}) + \sigma_t\epsilon_t \tag{14}$$

where $\hat{\mathbf{y}}_0$ is the denoised score in Equation 13. For detailed derivation, please refer to Appendix D. By substituting Equation 13 into Equation 14, we can obtain:

$$\begin{aligned}
\mathbf{y}_{t-1} = &\sqrt{\frac{\bar{\alpha}_{t-1}}{\bar{\alpha}_t}}(\mathbf{y}_t - (1 - \sqrt{\bar{\alpha}_t})g_\phi(\mathcal{E}, \mathbf{X}) - \sqrt{1 - \bar{\alpha}_t}\epsilon_\theta(\mathbf{y}_t, t, \mathcal{E}, \mathbf{X})) \\
&+ (1 - \sqrt{\bar{\alpha}_{t-1}})g_\phi(\mathcal{E}, \mathbf{X}) + \sqrt{1 - \bar{\alpha}_{t-1} - \sigma_t^2}\epsilon_\theta(\mathbf{y}_t, t, \mathcal{E}, \mathbf{X}) + \sigma_t\epsilon_t
\end{aligned} \tag{15}$$

This allows the use of a forward process defined only on a subset of the latent variables $\mathbf{y}_{\tau_1}, \ldots, \mathbf{y}_{\tau_t}$ where $\tau_1, \ldots, \tau_t$ is an increasing subsequence of $1, ..., T$ with length $S$, where $S$ could be much

smaller than $T$. To reduce the number of sampling steps from $T$ to $K$, we use $K$ evenly spaced real numbers between 1 and $T$ (inclusive), and then round each resulting number to the nearest integer, as follows: $\{\tau_i\}_{i=1}^{K} = \left\{ 1 + \frac{(T-1)(i-1)}{K-1} \right\}_{i=1}^{K}$.

When our prior is more confident, fewer sampling steps, or a smaller $K$, are needed, and vice versa. We propose a heuristic strategy to dynamically adjust the size of $K$ according to the confidence of different prior scores of anomalies. We choose the inverse sigmoid function to simulate the decay of the ratio as the confidence $|_\phi(\mathcal{E}, \mathbf{X}) - 0.5|$ increases:

$$K = \frac{r}{1 + \exp\left(\frac{|g_\phi(\mathcal{E}, \mathbf{X}) - 0.5|}{0.5}\right)} \times T \tag{16}$$

Typically, with $r$ set to 2, our framework adjusts the sampling steps $K$ to around 1000 for ambiguous priors near 0.5, and reduces it to about 500 for high-confidence priors close to 1. Notably, most nodes on the graph are associated with high prior confidence, which leads to a substantial decrease in computational demand. Conversely, for anomalous nodes that are adept at camouflage, the lower prior confidence necessitates a larger number of diffusion steps, facilitating their accurate detection. Our method thus strikes a balance between computational efficiency and thorough identification. We show the inference process with our prior-aware strided sampling in Appendix J.

## 5 Experiments

### 5.1 Experimental Setup

**Datasets**  We have extensively employed five diverse datasets from various domains to verify our method. They are the e-finance category dataset Elliptic [Weber et al., 2019], crowd-sourcing category datasets Tolokers [Platonov et al., 2023] and YelpChi [Rayana and Akoglu, 2015], and Social media datasets Question [Platonov et al., 2023] and Reddit [Kumar et al., 2019]. For the detail of dataset statistics and processing, please refer to Appendix G.

**Baselines**  We have compared our CGADM with two categories of methods in the context of graph anomaly detection: (1) Standard GNNs, which include GCN [Kipf and Welling, 2017], GIN [Xu et al., 2019], GraphSAGE [Hamilton et al., 2017], and GAT [Velickovic et al., 2018]; (2) GNNs specifically designed for anomaly detection, such as GAS [Li et al., 2019], PCGNN [Liu et al., 2021b], BWGNN [Tang et al., 2022], GHRN [Gao et al., 2023b], XGBGraph [Tang et al., 2023], and CONSISGAD [Chen et al., 2024]; (3) diffusion-based data-centric approaches for GAD: GODM [Ma et al., 2024a], CGenGA [Liu et al., 2023]. For detailed descriptions, please refer to Appendix E.

**Metrics**  Following the evaluation setup employed by most anomaly detection works [Han et al., 2022a], we have chosen the Area Under the Receiver Operating Characteristic Curve (AUROC) and the Area Under the Precision-Recall Curve (AUPRC) as our metrics for graph anomaly detection. Both of these metrics range between 0 and 1, and we record them as percentages for convenience. For both metrics, a higher value indicates better performance.

**Implementation Details**  For CGADM, the layer number of graph convolution is set to three, a value considered reasonable by most works [Liu et al., 2021b]. For our diffusion process, the noise levels at the initial and final time steps, $\beta_1$ and $\beta_T$, are set to 1e-4 and 0.02, respectively. Additionally, we employ linear interpolation to divide the time steps between them, which is consistent with DDPM [Ho et al., 2020]. For other implementation details, please refer to Appendix K.

### 5.2 Overall Comparison

We summarize the performance of all algorithms in terms of AUROC and AUPRC across different datasets in Table 1. We put more results of F1-score in Appendix N and results on additional datasets in Appendix A and L. The results demonstrate that our CGADM outperforms most other baselines across all metrics. We conduct two-sample t-tests, and $p - value < 0.05$ indicates that the improvements are statistically significant. In addition to these findings, we make the following observations:

Table 1: Performance Comparison on Graph Anomaly Detection

| Model | Ellip | | Tolo | | Yelp | | Quest | | Reddit | | **Average** | |
|---|---|---|---|---|---|---|---|---|---|---|---|---|
| | AUPRC | AUROC | AUPRC | AUROC | AUPRC | AUROC | AUPRC | AUROC | AUPRC | AUROC | AUPRC | AUROC |
| GCN | 80.19 | 95.12 | 41.44 | 73.58 | 23.59 | 59.89 | 10.27 | 67.73 | 5.65 | 62.55 | 32.23 | 71.77 |
| GIN | 83.88 | 96.21 | 37.89 | 74.02 | 38.13 | 77.40 | 11.23 | 68.07 | 5.38 | 65.25 | 35.30 | 76.19 |
| Graphsage | 86.16 | 96.61 | 43.73 | 77.30 | 50.23 | 83.24 | 13.86 | 70.64 | 5.78 | 63.67 | 39.95 | 78.29 |
| GAT | 87.59 | 97.11 | 42.18 | 76.66 | 46.64 | 80.95 | 13.19 | 68.19 | 5.42 | 63.55 | 39.00 | 77.29 |
| GAS | 87.54 | 97.14 | 42.39 | 74.55 | 39.18 | 78.63 | 12.41 | 66.09 | 5.66 | 61.23 | 37.44 | 75.53 |
| PCGNN | 67.29 | 93.88 | 36.76 | 71.28 | 45.32 | 79.61 | 13.79 | 69.12 | 4.13 | 54.58 | 33.46 | 73.69 |
| BWGNN | 87.90 | 96.99 | 45.02 | 77.80 | 49.15 | 81.85 | 14.64 | 69.96 | 5.42 | 60.63 | 40.43 | 77.45 |
| GHRN | 88.13 | 97.04 | 45.25 | 77.98 | 49.78 | 82.36 | 14.61 | 69.32 | 5.85 | 63.51 | 40.72 | 78.04 |
| XGBGraph | 90.47 | 94.35 | 44.47 | 77.28 | 75.91 | 91.85 | 14.33 | 64.90 | 4.59 | 60.58 | 45.95 | 77.79 |
| CONSISGAD | 86.42 | 96.38 | 40.59 | 76.03 | 41.74 | 79.35 | 12.85 | 70.54 | 5.57 | 66.99 | 37.43 | 77.86 |
| GODM | 85.89 | 93.92 | **46.15** | 76.42 | 51.77 | 84.33 | 15.11 | 68.86 | 5.55 | 62.10 | 40.89 | 77.13 |
| CGenGA | 87.36 | 96.07 | 44.89 | 78.95 | 52.76 | 85.65 | 15.34 | 68.46 | 5.78 | 64.78 | 41.23 | 78.78 |
| CGADM | **97.03** | **99.34** | 46.02 | **79.68** | **76.54** | **92.69** | **18.51** | 69.41 | 5.79 | 65.85 | **48.78** | **81.39** |

† **Boldface** denotes the highest score, and underline indicates the best result of the baselines.

- In terms of average performance, CGADM achieves 48.78% AUPRC and 81.39% AUROC, representing significant improvements of 6.15% in AUPRC and 4.53% in AUROC over the best baseline (XGBGraph for AUPRC and CONSISGAD for AUROC).

- GAD methods represent state-of-the-art methods. This indicates that GAD, with its unique challenges of data imbalance, data heterogeneity, and deliberate node obfuscation, cannot be adequately addressed by general GNNs and requires specialized design.

- No single baseline method consistently outperforms on all datasets. We believe this is because these discriminative models identify anomalous nodes through decision boundaries. Many anomalous nodes manage to cross these boundaries by obfuscating their features, making it difficult for these methods to adapt to various scenarios. In contrast, our CGADM consider the joint distribution of anomaly in a generative way, making it difficult for anomalous nodes to obfuscate.

- Diffusion-based approaches (GODM, CGenGA) that use data augmentation show competitive performance, but CGADM consistently outperforms them by directly modeling the joint anomaly distribution rather than relying on data augmentation techniques.

- Among standard GNN methods, GraphSage and GAT perform better than the other two methods, especially on the YelpChi dataset, which has significantly more edges. This aligns with our analysis in the introduction, where GNN, as a low-pass filter, blurs the distinctive features of anomalies in its inherent feature aggregation mechanism, a problem that worsens with an increased number of edges. GraphSage and GAT to some extent mitigate the over-smoothing issue by sampling neighbors or amplifying the weight of important neighbors, respectively.

## 5.3 Comparison with Different Prior Model

In generating the final anomaly value with CGADM, to ensure effectiveness, we do not start the reverse process from a random state. Instead, we opt for a conditional anomaly estimator to guide the reverse process of the model. For efficiency, we employ a lightweight ensemble trees model as the estimator. Here, we explore both Random Forest (RF) and Extreme Gradient Boosting Tree (XGBT) as estimator. We denote CGADM using RF and XGBT as conditional anomaly estimators as $CGADM_{RF}$ and $CGADM_{XGBT}$, respectively. Figure 2 records the performance of these models on the Elliptic and YelpChi datasets. Two observations can be made from figure 2. Firstly, both $CGADM_{RF}$ and $CGADM_{XGBT}$ outperform their corresponding initial priors. This proves that our CGADM's diffusion process can significantly enhance the performance of GAD. Secondly, the performance gap between $CGADM_{RF}$ and $CGADM_{XGBT}$ is significantly smaller than that between RF and XGBT. This indicates that our CGADM possesses strong robustness. Even in the face of initially inaccurate prior estimates, our CGADM can effectively correct the results under the iterative refinement of the topological-guided denoising network.

## 5.4 Parameter Sensitivity

**Impact of Graph Convolution Layer** $L$    In order to better capture the topological information surrounding nodes for joint distribution modeling, we employ a GNN-based encoder in our topological-guided denoising network. We explored the impact of the number of graph convolution layers on the

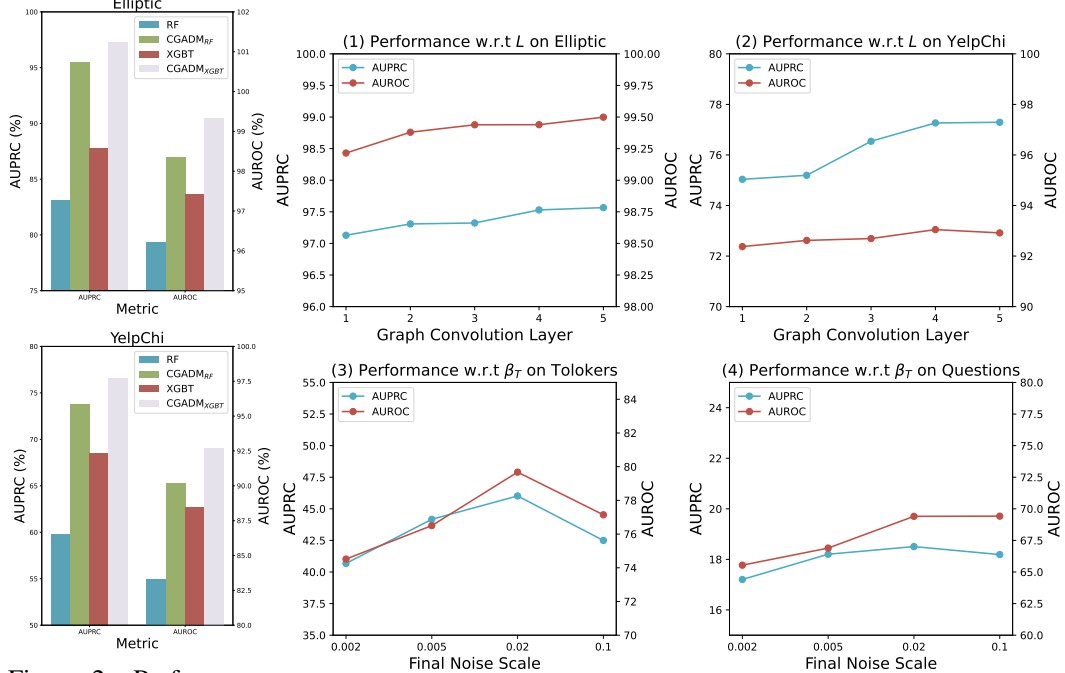

Figure 2: Performance w.r.t. Different Prior Models

Figure 3: Parameter Sensitivity on Different Datasets

Elliptic and YelpChi datasets. The results are shown in Figures 3 (1) and (2). From the results, we can observe a slowly gradual improvement in performance as the number of layers increases, reaching farther topological structure information. Even at a depth of five layers, there is no performance degradation. This suggests that our CGADM can effectively overcome the over-smoothing problem commonly encountered in traditional discriminative methods based on GNNs. We attribute this mainly to two factors. First, the paradigm shift to generating the joint distribution of anomaly on the graph allows considering the influence of surrounding neighbor nodes. Second, our residual propagation mechanism prevents the high-frequency information of nodes, thereby retaining more valuable information for anomaly value generation.

**Impact of the Final Noise Scale** $\beta_T$     We modify the endpoint of CGADM's diffusion process from the conventional Gaussian distribution $N(0, I)$ to $N(g_\phi(\mathcal{E}, \mathbf{X}), I)$. Intuitively, $\beta_T$ represents the maximum degree to which our noise-added $\mathbf{y}_t$ can deviate from the ground truth. It also represents the maximum scale at which our denoising network can correct the prior. We studied the magnitude of this degree on the Tolokers and Questions datasets, with the results shown in Figure 3 (3) and (4). We can observe that as the maximum correction scale increases, the performance initially improves. This suggests that the bias of the prior can be better corrected at this point. However, when the correction scale exceeds 0.02, the performance begins to decline as the maximum correction scale continues to increase. This may because the maximum correction scale has already surpassed the maximum bias produced by the prior. Overcorrection of the prior could prevent CGADM from modeling the true distribution. Therefore, we recommend using $\beta_T = 0.02$ in our cases,

## 5.5   Efficiency Analysis

In Section 4.4, we designed a prior-aware strided sampling strategy to adaptively reduce the reverse steps needed to generate anomaly values. To verify its efficiency, we designed the following two ablation experiments. In the first experiment, we tested the computation time and corresponding model performance of our CGADM with different sampling steps during generation. The results are shown in Figure 4. As can be seen, as our striding magnitude increases, i.e., the reverse steps of sampling become fewer, both computation time and model performance decrease. However, the decline in computation time is much greater than the decline in graph anomaly detection performance. Even when the striding is not large at the beginning, the decline in performance is not significant.

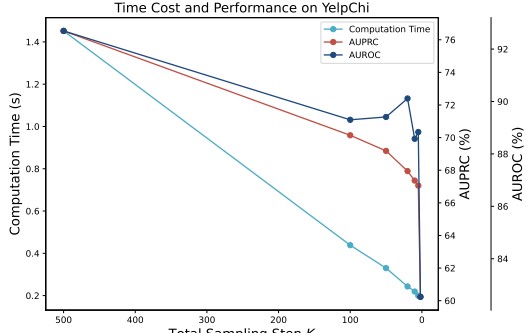

Figure 4: Time cost and Accuracy w.r.t. Sampling Steps $K$

| | CGADM | CGADM$_S$ |
|---|---|---|
| Average Reverse Step | 1000 | 583.0256 |
| AUPRC (%) | 76.5424 | 73.6636 |
| AUROC (%) | 92.6930 | 91.9423 |

Table 2: Performance Metrics

This implies that sacrificing a little performance can result in substantial savings in computation time. Therefore, we designed another ablation experiment. Here, we denote CGADM configured with prior-aware strided sampling as CGADM$_s$ and present its model performance and average reverse steps during inference in Table 2. Compared to the original 1000 sampling steps, our method reduces the average sampling steps for all nodes to 583, while ensuring only a slight drop in model performance, which remains highly competitive.

## 6   Conclusions and Limitation

Existing GNN-based graph anomaly detection methods are vulnerable to fraudulent nodes due to their feature aggregation and discriminative nature. To address this, we propose the Conditional Graph Anomaly Diffusion Model (CGADM), which considers node anomalies holistically across the graph, generating a distribution of anomaly values. We introduce a prior-guided diffusion process with a pre-trained conditional anomaly estimator to constrain the diffusion. Additionally, we implement a confidence-aware mechanism to adaptively determine reverse time steps, improving computational efficiency. Experimental results on standard benchmarks demonstrate that CGADM achieves state-of-the-art performance.

While CGADM shows strong performance, a few limitations remain. First, the model's reliance on pre-trained anomaly priors may require adaptation for applications with dynamic graph structures. Second, the current approach assumes a supervised setting, while real-world applications often require adaptation to unsupervised scenarios. These issues are areas for future improvement.

## Acknowledgments and Disclosure of Funding

This research was supported by the National Key R&D Program of China (No. 2023YFC3304701), NSFC (No.6250072448 and No.U24A20233), the Shandong Provincial Natural Science Foundation (No.ZQ2022JQ32), the Beijing Natural Science Foundation (L247027), the Fundamental Research Funds for the Central Universities, and the Research Funds of Renmin University of China. It was also supported by Big Data and Responsible Artificial Intelligence for National Governance, Renmin University of China.

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

## A  Evaluation on Additional Datasets

To further validate the generalizability of our approach, we conducted experiments on four additional real-world datasets Tang et al. [2023]: Weibo, Amazon, T-Finance, and T-Social. These datasets represent diverse application domains and vary in their structural properties and anomaly distributions. Tables 3 and 4 present the AUPRC and AUROC results, respectively, comparing our CGADM with state-of-the-art methods XGBGraph and CONSISGAD.

Table 3: AUPRC comparison on additional datasets

| Model | Weibo | Amazon | T-Finance | T-Social |
|---|---|---|---|---|
| XGBGraph | 0.9516 | 0.9020 | 0.8836 | 0.9203 |
| CONSISGAD | 0.8847 | 0.8047 | 0.7283 | 0.5212 |
| **CGADM (Ours)** | **0.9735** | **0.9191** | **0.9154** | **0.9408** |

Table 4: AUROC comparison on additional datasets

| Model | Weibo | Amazon | T-Finance | T-Social |
|---|---|---|---|---|
| XGBGraph | **0.9937** | 0.9682 | 0.9623 | **0.9914** |
| CONSISGAD | 0.9654 | 0.9409 | 0.9026 | 0.8963 |
| **CGADM (Ours)** | 0.9879 | **0.9736** | **0.9708** | 0.9761 |

The results demonstrate that CGADM consistently outperforms both XGBGraph and CONSISGAD in terms of AUPRC across all four additional datasets. While XGBGraph achieves marginally higher AUROC on Weibo and T-Social datasets, CGADM maintains competitive performance and excels on Amazon and T-Finance datasets. These comprehensive evaluations across nine diverse datasets underscore the robustness and effectiveness of our generative approach to graph anomaly detection across various domains and graph structures.

## B  Common Process of Diffusion Probabilistic Model

Here we show the common steps in the diffusion model as follows:

- **Forward process:** Given an input data sample $x_0 \sim q(x_0)$, the forward process constructs the latent variables $x_{1:T}$ in a Markov chain by progressively adding Gaussian noises over $T$ steps. Specifically, the forward transition $x_{t-1} \rightarrow x_t$ is defined as $q(x_t|x_{t-1}) = \mathcal{N}(x_t; \sqrt{1-\beta_t}x_{t-1}, \beta_t I)$, where $t \in \{1, ..., T\}$ refers to the diffusion step, $\mathcal{N}$ denotes the Gaussian distribution, and $\beta_t \in (0, 1)$ regulates the noise scales added at step $t$. If $T \rightarrow \infty$, $x_T$ approaches a standard Gaussian distribution [Ho et al., 2020].

- **Reverse process:** Diffusion models (DMs) aim to remove the added noises from $x_t$ to recover $x_{t-1}$ in the reverse step, striving to capture minor alterations in the complex generation process. Formally, taking $x_T$ as the initial state, DMs learn the denoising process $x_t \rightarrow x_{t-1}$ iteratively by $p_\theta(x_{t-1}|x_t) = \mathcal{N}(x_{t-1}; \mu_\theta(x_t, t), \Sigma_\theta(x_t, t))$, where $\mu_\theta(x_t, t)$ and $\Sigma_\theta(x_t, t)$ are the mean and covariance of the Gaussian distribution predicted by a neural network with parameters $\theta$.

- **Optimization:** DMs are optimized by maximizing the Evidence Lower Bound (ELBO) of the likelihood of observed input data $x_0$. Denote $\mathbb{D}_{KL}(p||q)$ as the Kullback–Leibler (KL) divergence from distribution $p$ to distribution $q$:

$$
\begin{aligned}
\log p(x_0) = \log \int p(x_{0:T}) dx_{1:T} &= \log \mathbb{E}_{q(x_{1:T}|x_0)} \left[ \frac{p(x_{0:T})}{q(x_{1:T}|x_0)} \right] \\
&\geq \mathbb{E}_{q(x_{1:T}|x_0)} \left[ \frac{p(x_{0:T})}{q(x_{1:T}|x_0)} \right] \\
&= \mathbb{E}_{q(x_1|x_0)} \left[ \log p_\theta(x_0|x_1) \right] - \mathbb{D}_{KL}(q(x_T|x_0)||p(x_T)) \\
&\quad - \sum_{t=2}^{T} \mathbb{E}_{q(x_t|x_0)} \left[ \mathbb{D}_{KL}(q(x_{t-1}|x_t, x_0)||p_\theta(x_{t-1}|x_t)) \right]
\end{aligned}
\tag{17}
$$

- **Inference:** After training $\theta$, DMs can draw $x_T \sim \mathcal{N}(0, I)$ and use $p_\theta(x_{t-1}|x_t)$ to iteratively repeat the generation process $x_T \to x_{T-1} \to \ldots \to x_0$.

## C   Posterior Coefficients Derivation

Similar to Han et al. [2022b], here we give the detailed derivation of Equation 6 and 7.

$$
\begin{aligned}
&q(\mathbf{y}_{t-1}|\mathbf{y}_t, \mathbf{y}_0, \mathcal{E}, \mathbf{X}) \\
&= q(\mathbf{y}_{t-1}|\mathbf{y}_t, \mathbf{y}_0, g_\phi(\mathcal{E}, \mathbf{X})) \propto q(\mathbf{y}_t|\mathbf{y}_{t-1}, g_\phi(\mathcal{E}, \mathbf{X})) q(\mathbf{y}_{t-1}|\mathbf{y}_0, g_\phi(\mathcal{E}, \mathbf{X})) \\
&\propto \exp\left( -\frac{1}{2}\left( \frac{\left(\mathbf{y}_t - \left(1 - \sqrt{\alpha_t}\right) g_\phi(\mathcal{E}, \mathbf{X}) - \sqrt{\alpha_t}\mathbf{y}_{t-1}\right)^2}{\beta_t} \right. \right. \\
&\left. \left. + \frac{\left(\mathbf{y}_{t-1} - \sqrt{\bar{\alpha}_{t-1}}\mathbf{y}_0 - (1 - \sqrt{\bar{\alpha}_{t-1}}) g_\phi(\mathcal{E}, \mathbf{X})\right)^2}{1 - \bar{\alpha}_{t-1}} \right) \right) \\
&\propto \exp\left( -\frac{1}{2}\left( \frac{\alpha_t \mathbf{y}_{t-1}^2 - 2\sqrt{\alpha_t}\left(\mathbf{y}_t - \left(1 - \sqrt{\alpha_t}\right) g_\phi(\mathcal{E}, \mathbf{X})\right)\mathbf{y}_{t-1}}{\beta_t} \right. \right. \\
&\left. \left. + \frac{\mathbf{y}_{t-1}^2 - 2\left(\sqrt{\bar{\alpha}_{t-1}}\mathbf{y}_0 + (1 - \sqrt{\bar{\alpha}_{t-1}}) g_\phi(\mathcal{E}, \mathbf{X})\right)\mathbf{y}_{t-1}}{1 - \bar{\alpha}_{t-1}} \right) \right) \\
&= \exp(-\frac{1}{2}((\underbrace{\frac{\alpha_t}{\beta_t} + \frac{1}{1 - \bar{\alpha}_{t-1}}}_{\text{Term 1}})\mathbf{y}_{t-1}^2 \\
&- 2(\underbrace{\frac{\sqrt{\bar{\alpha}_{t-1}}}{1 - \bar{\alpha}_{t-1}}\mathbf{y}_0 + \frac{\sqrt{\alpha_t}}{\beta_t}\mathbf{y}_t + \left(\frac{\sqrt{\alpha_t}\left(\sqrt{\alpha_t} - 1\right)}{\beta_t} + \frac{1 - \sqrt{\bar{\alpha}_{t-1}}}{1 - \bar{\alpha}_{t-1}}\right) g_\phi(\mathcal{E}, \mathbf{X})}_{\text{Term 2}})\mathbf{y}_{t-1})),
\end{aligned}
\tag{18}
$$

where

$$
\text{Term 1} = \frac{\alpha_t\left(1 - \bar{\alpha}_{t-1}\right) + \beta_t}{\beta_t\left(1 - \bar{\alpha}_{t-1}\right)} = \frac{1 - \bar{\alpha}_t}{\beta_t\left(1 - \bar{\alpha}_{t-1}\right)},
\tag{19}
$$

$$
\tilde{\beta}_t = \frac{1}{(1)} = \frac{1 - \bar{\alpha}_{t-1}}{1 - \bar{\alpha}_t}\beta_t,
\tag{20}
$$

Afterwards, we divide each coefficient in Term 2 by Term 1.

$$
\gamma_0 = \frac{\sqrt{\bar{\alpha}_{t-1}}}{1 - \bar{\alpha}_{t-1}}/1 = \frac{\sqrt{\bar{\alpha}_{t-1}}}{1 - \bar{\alpha}_t}\beta_t
\tag{21}
$$

$$
\gamma_1 = \frac{\sqrt{\alpha_t}}{\beta_t}/1 = \frac{1 - \bar{\alpha}_{t-1}}{1 - \bar{\alpha}_t}\sqrt{\alpha_t},
\tag{22}
$$

and

$$
\begin{aligned}
\gamma_2 &= \left(\frac{\sqrt{\alpha_t}\left(\sqrt{\alpha_t} - 1\right)}{\beta_t} + \frac{1 - \sqrt{\bar{\alpha}_{t-1}}}{1 - \bar{\alpha}_{t-1}}\right)/1 \\
&= \frac{\alpha_t - \bar{\alpha}_t - \sqrt{\alpha_t}\left(1 - \bar{\alpha}_{t-1}\right) + \beta_t - \beta_t\sqrt{\bar{\alpha}_{t-1}}}{1 - \bar{\alpha}_t} \\
&= 1 + \frac{\left(\sqrt{\bar{\alpha}_t} - 1\right)\left(\sqrt{\alpha_t} + \sqrt{\bar{\alpha}_{t-1}}\right)}{1 - \bar{\alpha}_t}.
\end{aligned}
\tag{23}
$$

Finally, we put every $\gamma_0$, $\gamma_1$, and $\gamma_2$ together and obtain Equation 6 and 7.

$$
\tilde{\mu}\left(\mathbf{y}_t, \mathbf{y}_0, g_\phi(\mathcal{E}, \mathbf{X})\right) = \gamma_0\mathbf{y}_0 + \gamma_1\mathbf{y}_t + \gamma_2 g_\phi(\mathcal{E}, \mathbf{X})
\tag{24}
$$

# D  Derivation of conditional non-Markovian reverse process

Following DDIM, we formally carry out the derivation of discarding the Markov constraint introduced by Equation 4 in our prior-conditional reverse step Equation 6. First, let's organize our target: given $q\left(\mathbf{y}_t \mid \mathbf{y}_0, g_\phi(\mathcal{E}, \mathbf{X})\right)$ and $q\left(\mathbf{y}_{t-1} \mid \mathbf{y}_0, g_\phi(\mathcal{E}, \mathbf{X})\right)$, without $q\left(\mathbf{y}_t \mid \mathbf{y}_{t-1}\right)$, we aim to find $q\left(\mathbf{y}_{t-1} \mid \mathbf{y}_t, \mathbf{y}_0, g_\phi(\mathcal{E}, \mathbf{X})\right)$.

Here we assume that $\mathbf{y}_{t-1}$ is a linear combination of $\mathbf{y}_t$, $\mathbf{y}_0$ and prior $g_\phi(\mathcal{E}, \mathbf{X})$ with coefficients denoted as $m_t$, $n_t$ and $o_t$, respectively. That is,

$$\mathbf{y}_{t-1} = m_t \mathbf{y}_t + n_t \mathbf{y}_0 + o_t g_\phi(\mathcal{E}, \mathbf{X}) + \sigma_t \epsilon_1 \tag{25}$$

We also know that

$$\mathbf{y}_t = \sqrt{\bar{\alpha}_t}\mathbf{y}_0 + (1 - \sqrt{\bar{\alpha}_t})g_\phi(\mathcal{E}, \mathbf{X}) + \sqrt{1 - \bar{\alpha}_t}\epsilon_2, \tag{26}$$

$$\mathbf{y}_{t-1} = \sqrt{\bar{\alpha}_{t-1}}\mathbf{y}_0 + (1 - \sqrt{\bar{\alpha}_{t-1}})g_\phi(\mathcal{E}, \mathbf{X}) + \sqrt{1 - \bar{\alpha}_{t-1}}\epsilon_3. \tag{27}$$

Here, the subscripts of $\epsilon_n$ are used to distinguish different samples from the Gaussian distribution. Substituting Equation 26 into Equation 25, we get

$$\mathbf{y}_{t-1} = m_t \left(\sqrt{\bar{\alpha}_t}\mathbf{y}_0 + (1 - \sqrt{\bar{\alpha}_t})g_\phi(\mathcal{E}, \mathbf{X}) + \sqrt{1 - \bar{\alpha}_t}\epsilon_2\right) + n_t \mathbf{y}_0 + o_t g_\phi(\mathcal{E}, \mathbf{X}) + \sigma_t \epsilon_1 \tag{28}$$

$$= \left(m_t\sqrt{\bar{\alpha}_t} + n_t\right)\mathbf{y}_0 + (m_t - m_t\sqrt{\bar{\alpha}_t} + o_t)g_\phi(\mathcal{E}, \mathbf{X}) + m_t\sqrt{1 - \bar{\alpha}_t}\epsilon_2 + \sigma_t \epsilon_1 \tag{29}$$

Therefore, we have

$$m_t\sqrt{\bar{\alpha}_t} + n_t = \sqrt{\bar{\alpha}_{t-1}}, \tag{30}$$

$$m_t^2\left(1 - \alpha_t\right) + \sigma_t^2 = 1 - \bar{\alpha}_{t-1}, \tag{31}$$

$$m_t - m_t\sqrt{\bar{\alpha}_t} + o_t = 1 - \sqrt{\bar{\alpha}_{t-1}} \tag{32}$$

Immediately, we can calculate $m_t$ and $n_t$:

$$m_t = \sqrt{\frac{1 - \bar{\alpha}_{t-1} - \sigma_t^2}{1 - \bar{\alpha}_t}}, \tag{33}$$

$$n_t = \sqrt{\bar{\alpha}_{t-1}} - \sqrt{\frac{\bar{\alpha}_t}{1 - \bar{\alpha}_t}\left(1 - \bar{\alpha}_{t-1} - \sigma_t^2\right)}, \tag{34}$$

$$o_t = 1 - \sqrt{\bar{\alpha}_{t-1}} - \sqrt{\frac{1 - \bar{\alpha}_{t-1} - \sigma_t^2}{1 - \bar{\alpha}_t}}(1 - \sqrt{\bar{\alpha}_t}). \tag{35}$$

Substituting back into Equation 25, we have

$$\mathbf{y}_{t-1} = \sqrt{\frac{1 - \bar{\alpha}_{t-1} - \sigma_t^2}{1 - \bar{\alpha}_t}}\mathbf{y}_t + \left(\sqrt{\bar{\alpha}_{t-1}} - \sqrt{\frac{\bar{\alpha}_t}{1 - \bar{\alpha}_t}\left(1 - \bar{\alpha}_{t-1} - \sigma_t^2\right)}\right)\mathbf{y}_0$$

$$+ (1 - \sqrt{\bar{\alpha}_{t-1}} - \sqrt{\frac{1 - \bar{\alpha}_{t-1} - \sigma_t^2}{1 - \bar{\alpha}_t}}(1 - \sqrt{\bar{\alpha}_t}))g_\phi(\mathcal{E}, \mathbf{X}) + \sigma_t \epsilon \tag{36}$$

$$= \sqrt{\bar{\alpha}_{t-1}}\mathbf{y}_0 + (1 - \sqrt{\bar{\alpha}_{t-1}})g_\phi(\mathcal{E}, \mathbf{X})$$

$$+ \sqrt{1 - \bar{\alpha}_{t-1} - \sigma_t^2}\left(\frac{1}{\sqrt{1 - \bar{\alpha}_t}}\mathbf{y}_t - \frac{\sqrt{\bar{\alpha}_t}}{\sqrt{1 - \bar{\alpha}_t}}\mathbf{y}_0 - \frac{1 - \sqrt{\bar{\alpha}_t}}{\sqrt{1 - \bar{\alpha}_t}}g_\phi(\mathcal{E}, \mathbf{X})\right) + \sigma_t \epsilon \tag{37}$$

$$= \sqrt{\bar{\alpha}_{t-1}}\mathbf{y}_0 + (1 - \sqrt{\bar{\alpha}_{t-1}})g_\phi(\mathcal{E}, \mathbf{X})$$

$$+ \sqrt{1 - \bar{\alpha}_{t-1} - \sigma_t^2}\frac{\mathbf{y}_t - \sqrt{\bar{\alpha}_t}\mathbf{y}_0 - (1 - \sqrt{\bar{\alpha}_t})g_\phi(\mathcal{E}, \mathbf{X})}{\sqrt{1 - \bar{\alpha}_t}} + \sigma_t \epsilon \tag{38}$$

Substituting the model's predicted value, we have

$$\mathbf{y}_{t-1} = \sqrt{\bar{\alpha}_{t-1}}\hat{\mathbf{y}}_{0|t} + (1 - \sqrt{\bar{\alpha}_{t-1}})g_\phi(\mathcal{E}, \mathbf{X}) + \sqrt{1 - \bar{\alpha}_{t-1} - \sigma_t^2}\epsilon_\theta(\mathbf{y}_t, t, \mathcal{E}, \mathbf{X}) + \sigma_t \epsilon \tag{39}$$

At this point, the derived result Equation 39 is completely consistent with Equation 15. That is, we use the two conditions $q\left(\mathbf{y}_t \mid \mathbf{y}_0, g_\phi(\mathcal{E}, \mathbf{X})\right)$ and $q\left(\mathbf{y}_{t-1} \mid \mathbf{y}_0, g_\phi(\mathcal{E}, \mathbf{X})\right)$, without $q\left(\mathbf{y}_t \mid \mathbf{y}_{t-1}\right)$, and obtain $q\left(\mathbf{y}_{t-1} \mid \mathbf{y}_t, \mathbf{y}_0, g_\phi(\mathcal{E}, \mathbf{X})\right)$. DDPM removes the condition $q\left(\mathbf{y}_t \mid \mathbf{y}_{t-1}\right)$, leading to the more general DDIM sampling formula.

# E  Baselines

In this section, we introduce the baseline models, which can be broadly bifurcated into two categories: (1) General-purpose graph neural networks, and (2) Techniques specifically designed for graph anomaly detection, and (3) Diffusion-based data augmentation approaches for graph anomaly detection. We have annotated each model with their respective categories for easy differentiation.

- **GCN** [Kipf and Welling, 2017] (1): This technique employs the convolution operation on graphs to propagate information from a node to its adjacent nodes. This allows the network to learn a representation for each node, grounded on its local neighborhood.

- **GIN** [Xu et al., 2019] (1): A variant of GNN, GIN is designed to encapsulate the graph's structure while maintaining graph isomorphism. This implies that it yields identical embeddings for graphs that are structurally indistinguishable, irrespective of permutations in their node labels.

- **GraphSAGE** [Hamilton et al., 2017] (1): This is an inductive learning framework that generates node embeddings by sampling and aggregating features from a node's local neighborhood.

- **GAT** [Velickovic et al., 2018] (1): This GNN framework incorporates the attention mechanism, assigning varying degrees of importance to different nodes during the neighborhood information aggregation process. This enables the model to concentrate on the most informative neighbors.

- **GAS** [Li et al., 2019] (2): This is a highly scalable technique for detecting spam reviews. It expands GCN to manage heterogeneous and heterophilic graphs and adapts to the graph structure of specific GAD applications using the KNN algorithm.

- **PCGNN** [Liu et al., 2021b] (2): This framework is designed for imbalanced GNN learning in fraud detection. It employs a label-balanced sampler to select nodes and edges for training, leading to a balanced label distribution in the induced sub-graph. Additionally, it uses a learnable parameterized distance function to select neighbors, filtering out superfluous links and incorporating beneficial ones for fraud prediction.

- **BWGNN** [Tang et al., 2022] (2): This technique is proposed to address the 'right-shift' phenomenon of graph anomalies, where the spectral energy distribution focuses less on low frequencies and more on high frequencies. It utilizes the Beta kernel to tackle higher frequency anomalies through multiple flexible, spatial/spectral-localized, and band-pass filters.

- **GHRN** [Gao et al., 2023b] (2): This approach addresses the heterophily issue in the spectral domain of graph anomaly detection by pruning inter-class edges to highlight and outline the graph's high-frequency components.

- **XGBGraph** [Tang et al., 2023] (2): A gradient boosting framework that combines traditional XGBoost with graph structural features.

- **CONSISGAD** [Chen et al., 2024] (2): A consistency training approach that leverages learnable data augmentation for graph anomaly detection with limited supervision.

- **GODM** [Ma et al., 2024a] (3): A data-centric approach for graph anomaly detection with few labels. It employs a diffusion model to generate positive examples in the latent space, addressing the label imbalance problem that is inherent in anomaly detection tasks.

- **CGenGA** [Liu et al., 2023] (3): A framework that uses latent diffusion models for data augmentation in graph anomaly detection. It generates synthetic graph data to enhance the training of supervised outlier detection methods, particularly effective in scenarios with limited labeled anomalies.

## F  Challenge of Graph Anomaly Detection

Although GAD is essentially a binary node classification problem, it presents several unique challenges. Firstly, anomalous nodes typically constitute a small fraction of the total nodes, leading to a significant data imbalance [Liu et al., 2021b]. Secondly, graphs containing anomalies often exhibit strong heterophily, where connected nodes possess diverse features and labels [Gao et al., 2023b, Tang et al., 2023]. This heterophily necessitates the development of methods that can effectively handle neighborhood feature disparities during message passing. Lastly, anomalous nodes tend to camouflage their features and connections, striving to blend in by mimicking normal patterns within the graph [Liu et al., 2020].

## G  Details of the datasets

The detailed statistics of the datasets we used are in Table 5. In line with the data characteristics of anomaly detection, the selected datasets each contain over 100 anomaly points, and the proportion of anomalies does not exceed 25%, satisfying the inherent imbalance problem in graph anomaly detection [Tang et al., 2023]. For each dataset, we randomly selected 20% of the points as training data, 10% of the points as validation data, and the remaining points as test data.

Table 5: Descriptive statistics of the datasets.

|  | #Nodes | #Edges | Feature Dim | Anomaly Ratio | Feature Type |
|---|---|---|---|---|---|
| Elliptic | 203,769 | 234,355 | 166 | 9.8% | Timestamps and transaction information |
| Tolokers | 11,758 | 519,000 | 10 | 21.8% | User profile with task performance statistics |
| YelpChi | 45,954 | 3,846,979 | 32 | 14.5% | Hand-crafted review features and statistics |
| Questions | 48,921 | 153,540 | 301 | 3.0% | FastText embeddings for user descriptions |
| Reddit | 10,984 | 168,016 | 64 | 3.3% | Hand-crafted review features and statistics |

## H  Implementation of Topological-guided Denoising Network

Reflecting upon Equation 9, we initially extend the formula of graph convolution to matrix form to facilitate computation across the entire graph, as shown below:

$$\mathbf{H}^l = \sigma(\mathbf{W}^{l-1}(\mathbf{I} - \mathbf{D}^{-1}\mathbf{A}\mathbf{H}^{l-1}))$$

After conducting $L$ rounds of convolution, we use weighted summation as our aggregation function for the hidden representations obtained from each layer of graph convolution. The formula is as follows:

$$\mathbf{H}^{final} = AGG(\mathbf{H}^1, \mathbf{H}^2, \ldots, \mathbf{H}^L) = \sum_{l=0}^{L} \alpha_l \mathbf{H}^l$$

Here, $\alpha_l$ are the weights for each layer's representation, which can be learned during training. Having obtained the representation of nodes that integrates both topological structure and node features, we construct our denoising function $\epsilon_\theta(\mathbf{y}_t, t, \mathbf{H}^{final})$ through a Multilayer Perceptron (MLP). Following the original DDPM Ho et al. [2020], we also adopt position embedding to encode time $t$. Therefore, the denoising function $\epsilon_\theta$ is as follows:

$$\epsilon_\theta = MLP(\text{Concat}[Pos(\mathbf{t}), \mathbf{y}_t, \mathbf{H}^{final}])$$

In this equation, $Pos(\mathbf{t})$ represents the position embedding of time $mathbft$, $\mathbf{y}_t$ is the current representation of the nodes, and $\mathbf{H}^{final}$ is the final aggregated representation after $L$ layers of graph convolution.

## I  Training of CGADM

According to the loss in Equation 11, the pseudo algorithm for training is shown in Algorithm 2

---

**Algorithm 2** CGADM Training

---

1: Pre-train $g_\phi(\mathcal{E}, \mathbf{X})$ that predicts the anomaly prior
2: **repeat**
3:  Draw $\mathbf{t} \sim \text{Uniform}(\{1, \dots, T\})$
4:  Draw $\epsilon \sim \mathcal{N}(0, I)$
5:  Compute the noise estimation loss:

$$\mathcal{L}_\epsilon = ||\epsilon - \epsilon_\theta(\sqrt{\bar{\alpha}_t}\mathbf{y}_0 + (1 - \sqrt{\bar{\alpha}_t})g_\phi(\mathcal{E}, \mathbf{X}) + \sqrt{1 - \bar{\alpha}_t}\epsilon, t, \mathcal{E}, \mathbf{X})||^2$$

6:  Take a numerical optimization step on $\nabla_\theta L_\epsilon$
7: **until** Convergence

---

## J  Inference with Prior-aware Strided Sampling

We show the complete pseudo algorithm for inference with our prior-aware strided sampling strategy in Algorithm 3

---

**Algorithm 3** Inference for Anomaly Detection with Sampling Strategy

---

1: Initialize $\mathbf{y}_T \sim \mathcal{N}(g_\phi(\mathcal{E}, \mathbf{X}), I)$
2: Compute $K$ based on the prior confidence $|g_\phi(\mathcal{E}, \mathbf{X}) - 0.5|$ using:

$$K = \frac{r}{1 + \exp\left(\frac{|g_\phi(\mathcal{E}, \mathbf{X}) - 0.5|}{0.5}\right)} \times T$$

  where $r$ is a hyperparameter.
3: Generate sampling time steps $\{\tau_i\}_{i=1}^K$:

$$\tau_i = \left\lfloor 1 + \frac{(T-1)(i-1)}{K-1} \right\rfloor, \quad i = 1, \dots, K$$

4: **for** $i = K$ to $1$ **do**
5:  Set $t = \tau_i$
6:  Calculate reparameterized $\hat{\mathbf{y}}_0$ using Equation 13:

$$\hat{\mathbf{y}}_0 = \frac{1}{\sqrt{\bar{\alpha}_t}}\left(\mathbf{y}_t - (1 - \sqrt{\bar{\alpha}_t})g_\phi(\mathcal{E}, \mathbf{X}) - \sqrt{1 - \bar{\alpha}_t}\epsilon_\theta(\mathbf{y}_t, t, \mathcal{E}, \mathbf{X})\right)$$

7:  **if** $i > 1$ **then**
8:   Draw $z \sim \mathcal{N}(0, I)$
9:   Update $\mathbf{y}_{t-1}$ using the modified non-Markovian reverse process:

$$\mathbf{y}_{t-1} = \sqrt{\bar{\alpha}_{\tau_{i-1}}}\hat{\mathbf{y}}_0 + (1 - \sqrt{\bar{\alpha}_{\tau_{i-1}}})g_\phi(\mathcal{E}, \mathbf{X}) + \sqrt{1 - \bar{\alpha}_{\tau_{i-1}} - \sigma_t^2}\epsilon_\theta(\mathbf{y}_t, t, \mathcal{E}, \mathbf{X}) + \sigma_t z$$

10:  **else**
11:   Set $\mathbf{y}_{t-1} = \hat{\mathbf{y}}_0$
12:  **end if**
13: **end for**
14: **return** $y_0$

---

## K  Implementation Detail

All experiments were conducted on a Linux machine equipped with an Nvidia GeForce RTX 3090. The CUDA version used was 11.1, and the driver version was 455.45.01. We implemented our algorithm and the corresponding baseline methods using PyTorch [Paszke et al., 2019] and the graph computation framework Pytorch-Geometric [Fey and Lenssen, 2019]. For the Random Forest (RF) and Extreme Gradient Boosting Tree (XGBT) that serve as conditional anomaly estimators, we used the RF version implemented in the Scikit-Learn library Pedregosa et al. [2011]. For XGBoost Chen and Guestrin [2016], we utilized its official implementation.

Table 6: Performance comparison on the DGraph dataset.

| Method | AUPRC | AUROC |
|---|---|---|
| GCN | 3.66 | 74.97 |
| GIN | 3.22 | 73.14 |
| GraphSAGE | 3.43 | 73.81 |
| GAT | 3.65 | 75.17 |
| GAS | 2.91 | 71.21 |
| PCGNN | 2.82 | 71.78 |
| BWGNN | 3.63 | 75.16 |
| GHRN | 3.68 | 75.15 |
| CGADM | **3.83** | **76.43** |

We initialize the latent vectors for all models with a Gaussian Distribution, having a mean value of 0 and a standard deviation of 0.01. To ensure a level playing field, the dimension of the hidden layer for all baseline models, as well as our CGADM, is set to 64. We conducted a grid search for hyper-parameter tuning. The learning rates were selected from the set [0.005, 0.01, 0.02, 0.05]. To prevent overfitting, we incorporated an L2 norm with the coefficient tuned from the set [0.001, 0.005, 0.01, 0.02, 0.1]. For all methods, we selected the best models by implementing early stopping when the AUROC on the validation set did not increase for five consecutive epochs.

## L  Efficacy in Highly Imbalanced Scenarios

We conducted additional experiments on the **DGraph** dataset Huang et al. [2022], a highly imbalanced real-world financial fraud detection dataset where anomalies constitute only **1.3%** of the data. The results are presented in Table 6:

As Table 6 illustrates, CGADM consistently outperforms all baseline methods on both AUPRC and AUROC metrics in this **extremely imbalanced setting**. Notably, the AUPRC metric demonstrates CGADM's ability to handle rare event detection by excelling in anomaly-specific precision and recall. Similarly, the superior AUROC indicates robust overall discriminative performance.

## M  Empirical Results on Efficiency

We conducted experiments to compare memory usage, training time, and inference time with baselines specifically designed for anomaly detection on the *Elliptic* dataset, which contains 203,769 nodes and 234,355 edges. The results are summarized in Table 7:

Table 7: Efficiency comparison on the Elliptic dataset.

| Model | Memory (MB) | Training Time (s/epoch) | Inference Time (s) |
|---|---|---|---|
| GAS | 1418 | 14.96 | 2.3865 |
| PCGN | 914 | 1.86 | 0.0827 |
| BWGNN | 446 | 0.75 | 0.1185 |
| GHRN | 924 | 1.57 | 0.1249 |
| **CGADM (ours)** | **1048** | **2.21** | **0.5691** |

From these empirical results, we draw the following observations:

- **Memory Efficiency:** The use of sparse matrix computations ensures that CGADM remains efficient in terms of memory usage, even for large-scale graphs. The marginal increase in memory usage is negligible compared to the scalability benefits.

- **Training Efficiency:** While CGADM's training time is moderately higher than discriminative methods (2.21s vs 0.75s for BWGNN), the performance gains (+10% AUPRC improvement over BWGNN) justify this reasonable overhead, especially considering the substantial improvement in detection capability.

- **Inference Time:** While our inference time is higher than most discriminative methods, the increase is justified given the novel generative anomaly detection paradigm. Considering the already low baseline inference time of anomaly detection tasks, the additional time overhead is acceptable, especially in scenarios where performance improvements are critical.

Overall, these results demonstrate that CGADM achieves state-of-the-art detection performance with reasonable computational demands, striking an effective balance between accuracy and efficiency. The slightly higher computational cost compared to discriminative methods is a worthwhile trade-off given the substantial performance improvements observed in our experiments.

# N    Additional Experiment Results

We computed the **F1-scores** for our model and baseline methods across all datasets. These results further confirm the superior performance of our model. Table 8 presents the F1-scores, which show consistency with the experiment results in Table 1.

Table 8: F1-scores comparison across datasets.

| Model | Ellip | Tolo | Yelp | Quest | Reddit |
|---|---|---|---|---|---|
| GCN | 73.672 | 47.376 | 27.658 | 6.856 | 7.794 |
| GIN | 75.338 | 49.443 | 42.214 | 10.288 | 6.443 |
| GraphSAGE | 81.096 | 50.226 | 43.949 | 12.041 | **10.075** |
| GAT | 80.498 | 50.878 | 48.891 | 11.157 | 8.432 |
| GAS | 77.844 | 48.253 | 43.404 | 10.867 | 9.071 |
| PCGNN | 45.090 | 47.213 | 44.608 | 5.796 | 6.981 |
| BWGNN | 83.134 | 49.983 | 47.323 | 12.788 | 6.501 |
| GHRN | 85.678 | 51.493 | 45.970 | 12.696 | 6.702 |
| XGBGraph | 87.555 | 51.079 | 65.121 | 16.088 | 2.954 |
| CONSISGAD | 79.120 | 49.762 | 41.606 | 9.848 | 6.443 |
| Ours (CGADM) | **93.390** | **51.595** | **69.396** | **17.162** | 9.754 |

# O    Robustness of CGADM against Feature Manipulation

To evaluate the robustness of CGADM against feature manipulation, we introduced feature perturbations in the **Elliptic** and **Tolokers** datasets. Specifically, we randomly perturbed the features of nodes with varying proportions (10%, 20%, and 30%) by randomly selecting values from their possible ranges with uniform probability. We then compared the performance of CGADM with GHRN (the best-performing baseline from our original experiments) under these conditions.

The results are summarized in Figure 5. As the proportion of perturbed nodes increases, the performance of both models decreases. However, CGADM consistently exhibits a slower decline compared to GHRN. This highlights CGADM's superior robustness to feature perturbations, which we attribute to its denoising reconstruction mechanism. This mechanism leverages information from neighboring nodes during the reverse diffusion process to iteratively restore the true anomaly signals.

# P    Effect of High- and Low-frequency Signals

To further substantiate that the high-frequency components are indeed reflected in the residual propagations, we designed an ablation study comparing our original CGADM (denoted as $CGADM_{HP}$) with a variant (denoted as $CGADM_{LP}$) that only propagates low-frequency signals. In $CGADM_{LP}$, the graph convolution operation is replaced with the standard GCN:

$$\frac{1}{|\mathcal{N}(v)| + 1} \left( \mathbf{h}_v^{l-1} + \sum_{u \in \mathcal{N}(v)} \mathbf{h}_u^{l-1} \right), \tag{40}$$

where the feature representation is averaged across the node and its neighbors, propagating only low-frequency signals.

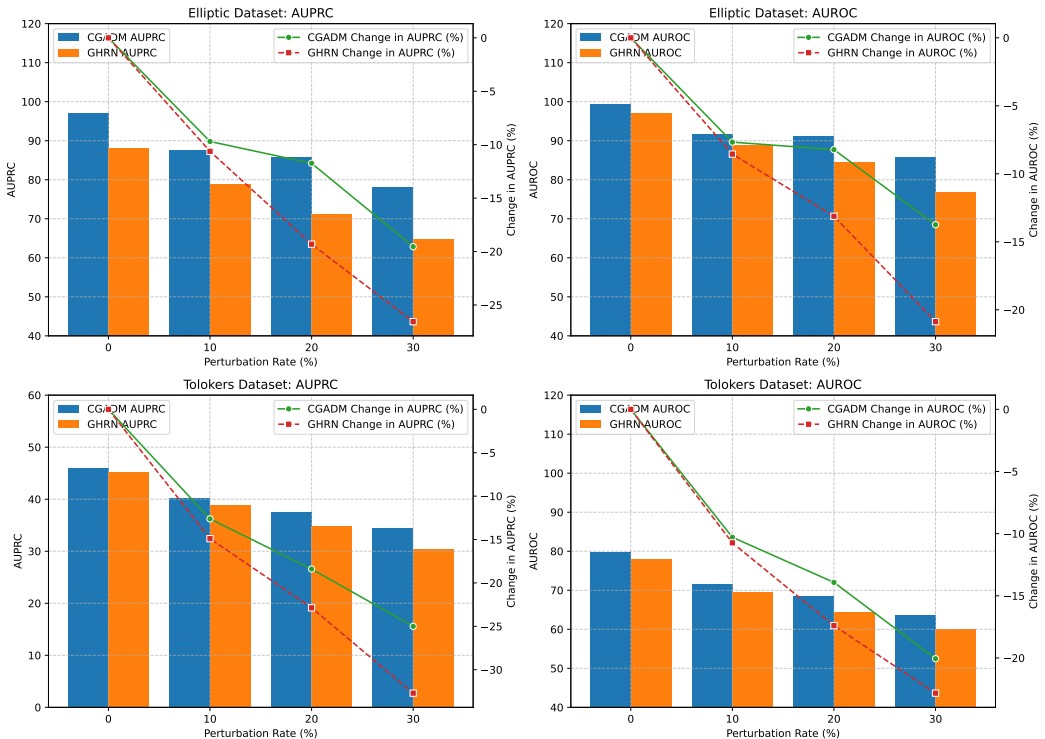

Figure 5: Robustness against Feature Manipulation

We conducted experiments on the *Elliptic* and *YelpChi* datasets, varying the number of GNN layers in the denoiser module. The results are shown in Table 9:

Table 9: Performance comparison of $CGADM_{HP}$ and $CGADM_{LP}$ with varying GNN layers.

| GNN Layers | Model | AUPRC (Elliptic) % | AUROC (Elliptic) % | AUPRC (YelpChi) % | AUROC (YelpChi) % |
|---|---|---|---|---|---|
| 1 | $CGADM_{HP}$ | 97.13 | 99.22 | 75.04 | 92.37 |
| | $CGADM_{LP}$ | 95.71 | 98.43 | 72.23 | 91.88 |
| 2 | $CGADM_{HP}$ | 97.31 | 99.38 | 75.20 | 92.62 |
| | $CGADM_{LP}$ | 93.73 | 97.60 | 70.92 | 90.88 |
| 3 | $CGADM_{HP}$ | 97.32 | 99.44 | 76.54 | 92.69 |
| | $CGADM_{LP}$ | 90.83 | 95.58 | 71.43 | 89.64 |
| 4 | $CGADM_{HP}$ | 97.53 | 99.44 | 77.27 | 93.05 |
| | $CGADM_{LP}$ | 87.12 | 92.60 | 69.98 | 87.71 |
| 5 | $CGADM_{HP}$ | 97.57 | 99.50 | 77.29 | 92.92 |
| | $CGADM_{LP}$ | 81.20 | 89.49 | 68.71 | 86.08 |

According to Table 9, we have the following observations:

1. **High-Frequency Signal Preservation Matters:** $CGADM_{HP}$, which retains high-frequency signals through residual propagation, consistently outperforms $CGADM_{LP}$ across all metrics and datasets. This highlights the importance of preserving high-frequency information for anomaly detection, as anomalies often manifest as local deviations that are captured by these components.

2. **Sensitivity to GNN Layers:** For $CGADM_{LP}$, performance declines significantly as the number of GNN layers increases. This is indicative of the well-known over-smoothing issue, where stacking multiple low-pass filters causes node representations to converge, losing discriminative information. Conversely, $CGADM_{HP}$ remains robust, and its performance even improves slightly with additional layers, demonstrating the effectiveness of residual propagation in mitigating over-smoothing.

3. **Iterative Refinement Amplifies Over-Smoothing:** In the context of our diffusion model, the iterative refinement process repeatedly aggregates neighborhood information, exacer-

bating the impact of over-smoothing in $CGADM_{LP}$. This leads to a failure to capture new anomaly-relevant signals at each stage of refinement. In contrast, $CGADM_{HP}$ avoids this issue by leveraging high-frequency signals to refine anomaly detection throughout the iterative process.

## Q    More Comparison with Data-augmentation Methods

The main distinction between CGADM and the existing data-augmentation methods lies in the underlying approach to anomaly detection. While prior works focus on using diffusion models for **data augmentation** to improve detection performance, CGADM adopts a **generative, model-centric paradigm** to directly model the joint distribution of anomalies on the entire graph. Below, we summarize the key differences:

- CAGAD [Xiao et al., 2024]: Uses a graph-specific diffusion model to generate counterfactual representations by transforming normal neighbors into anomalous ones. This is a classic **data augmentation** technique to enhance anomaly distinguishability.

- DEGAD [Pang et al., 2024]: Employs diffusion models to generate manipulated neighbors, enhancing graphs by creating augmented data. This technique is used as a **data enhancement module** within a contrastive learning framework.

- ConGNN [Li et al., 2024]: Introduces a generator based on diffusion models to control neighborhood aggregation and **create augmented data** for better anomaly detection performance.

- GD [Liu et al., 2024]: Tackles the label imbalance problem by **generating positive examples** using a diffusion model in the latent space. The primary goal is to balance datasets, not directly detect anomalies.

- Diffad [Ma et al., 2024b]: Investigates denoising diffusion models to **synthesize graph structures** and enhance existing methods. This approach focuses on data synthesis rather than directly detecting anomalies.

We have conducted a detailed experimental comparison of our proposed Conditional Graph Anomaly Diffusion Model (CGADM) with some diffusion-based data augmentation methods CAGAD [Xiao et al., 2024], DEGAD [Pang et al., 2024], ConGNN [Li et al., 2024], GD [Liu et al., 2024], and Diffad [Ma et al., 2024b]. We analyzed their performance across several standard benchmark datasets (Elliptic, Tolokers, and YelpChi), and the key results are summarized below:

Table 10: AUPRC and AUROC comparison with Data Augmentation Methods

| Metric | Model | Ellip | Tolo | Yelp |
|--------|-------|-------|------|------|
|        | CAGAD | 89.75 | 40.80 | 72.30 |
|        | DEGAD | 93.86 | 43.51 | 75.11 |
| AUPRC  | ConGNN | 91.60 | 42.22 | 73.60 |
|        | GD | 88.63 | 39.90 | 68.01 |
|        | Diffad | 90.05 | 41.75 | 71.28 |
|        | **CGADM** | **97.28** | **45.11** | **76.54** |
|        | CAGAD | 94.82 | 72.22 | 90.34 |
|        | DEGAD | 97.88 | 76.20 | 92.22 |
| AUROC  | ConGNN | 95.60 | 74.56 | 91.33 |
|        | GD | 93.53 | 70.70 | 83.84 |
|        | Diffad | 92.72 | 73.31 | 88.21 |
|        | **CGADM** | **99.34** | **78.11** | **92.69** |

As shown in the Table 10, CGADM consistently outperforms the data-augmentation methods in both AUPRC and AUROC across all datasets. This underscores the efficacy of our generative framework in addressing graph anomaly detection challenges.

## Q.1 Quantitative Analysis of Over-smoothing Mitigation

To provide quantitative evidence that CGADM effectively mitigates the over-smoothing problem, we conducted a Dirichlet Energy analysis, which measures the preservation of high-frequency signals in node embeddings. Dirichlet Energy is defined as:

$$E(f) = \frac{1}{2} \sum_{(i,j) \in E} w_{ij}(f(i) - f(j))^2$$

where $w_{ij}$ represents the weight of edge $(i, j)$, and $f(i)$ is the value of the embedding at node $i$. Higher Dirichlet Energy indicates better preservation of high-frequency signals, which is critical for distinguishing anomalous nodes.

We compared our CGADM with a variant where the GNN layers were replaced with traditional GCN layers, and the results are presented in Table 11.

Table 11: Dirichlet Energy comparison between CGADM and GCN-based variant

| Model | Dirichlet Energy (Elliptic) | Dirichlet Energy (Tolo) |
|---|---|---|
| CGADM | 105,002 | 3,977 |
| CGADM with GCN | 66,345 | 1,383 |

The results demonstrate that CGADM consistently produces embeddings with significantly higher Dirichlet Energy compared to the GCN-based variant across both datasets. This confirms that our residual propagation mechanism effectively preserves high-frequency signals that are critical for anomaly detection, thereby mitigating the over-smoothing problem common in traditional GNN approaches.

These findings complement our ablation studies in Section 5.4, where we showed that CGADM's performance improves with deeper GNN layers, and our analysis in Appendix P, which demonstrates the importance of preserving high-frequency components for effective anomaly detection.

# R  Theoretical Analysis of Over-smoothing Mitigation in CGADM

In this section, we provide a rigorous theoretical analysis of how our Conditional Graph Anomaly Diffusion Model (CGADM) effectively mitigates the over-smoothing problem typically encountered in deep GNNs while still capturing long-range dependencies.

## R.1  Background: Over-smoothing in GNNs

Over-smoothing in GNNs occurs when node representations become increasingly similar as more layers are stacked, eventually converging to indistinguishable representations. For a standard GNN with $L$ layers, the representation of a node $v$ at layer $l$ can be expressed as:

$$\mathbf{h}_v^{(l)} = \sigma \left( \mathbf{W}^{(l-1)} \sum_{u \in \mathcal{N}(v) \cup \{v\}} \frac{1}{|\mathcal{N}(v)| + 1} \mathbf{h}_u^{(l-1)} \right) \tag{41}$$

It has been shown that as $L \to \infty$, all node representations converge: $\|\mathbf{h}_v^{(L)} - \mathbf{h}_u^{(L)}\| \to 0$ for any nodes $v$ and $u$ in a connected graph.

## R.2  Receptive Field Analysis

We define the receptive field $\mathcal{R}_L(v)$ of a node $v$ after $L$ layers of message passing as the set of nodes whose features contribute to the final representation of $v$:

$$\mathcal{R}_L(v) = \{u \in \mathcal{V} \mid \text{dist}(u, v) \leq L\} \tag{42}$$

where $\text{dist}(u, v)$ represents the shortest path distance between nodes $u$ and $v$.

**Theorem R.1.** *For a CGADM model with an $L$-layer GNN denoiser and $T$ denoising steps, the effective receptive field of a node $v$ is $\mathcal{R}_{CGADM}^T(v) = \mathcal{R}_{L \times T}(v)$, equivalent to an $(L \times T)$-layer traditional GNN without the over-smoothing effect.*

*Proof.* In CGADM, each denoising step $t$ applies an $L$-layer GNN to refine the node representations. The key difference from traditional GNNs is our residual propagation mechanism in Equation (9):

$$\mathbf{h}_v^l = \sigma \left( \mathbf{W}^{l-1} \left( \mathbf{h}_v^{l-1} - \frac{1}{|\mathcal{N}(v)|} \sum_{u \in \mathcal{N}(v)} \mathbf{h}_u^{l-1} \right) \right) \tag{43}$$

For each denoising step $t$, we define the influence set $\mathcal{I}_v^{t \times L}$ as the set of nodes that contribute to the representation of node $v$ after $t$ denoising steps, each involving $L$ graph convolution layers.

For $t = 1$, the influence set is identical to the receptive field of an $L$-layer GNN:

$$\mathcal{I}_v^{1 \times L} = \mathcal{R}_L(v) \tag{44}$$

For successive denoising steps, the influence set expands recursively:

$$\mathcal{I}_v^{t \times L} = \bigcup_{u \in \mathcal{I}_v^{(t-1) \times L}} \mathcal{R}_L(u) \tag{45}$$

This recursive expansion leads to:

$$\mathcal{I}_v^{T \times L} = \mathcal{R}_{L \times T}(v) \tag{46}$$

Thus, after $T$ denoising steps, the effective receptive field of node $v$ in CGADM encompasses nodes up to $L \times T$ hops away, equivalent to an $(L \times T)$-layer traditional GNN.

To prove that over-smoothing is mitigated, we analyze the residual propagation mechanism. Unlike standard GNNs that apply a low-pass filter by averaging features, our approach computes the difference between the node's feature and the average of its neighbors' features:

$$\mathbf{h}_v^l - \frac{1}{|\mathcal{N}(v)|} \sum_{u \in \mathcal{N}(v)} \mathbf{h}_u^{l-1} \tag{47}$$

This operation is equivalent to a high-pass filter that preserves the high-frequency components of the signal. In the spectral domain, for a graph signal $\mathbf{x}$ with Fourier coefficients $\hat{\mathbf{x}}$, the residual propagation applies a transfer function:

$$H(\lambda_i) = 1 - \lambda_i \tag{48}$$

where $\lambda_i$ are the eigenvalues of the normalized Laplacian matrix. This transfer function amplifies the contribution of eigenvectors corresponding to larger eigenvalues (high-frequency components) while reducing the contribution of eigenvectors corresponding to smaller eigenvalues (low-frequency components).

Consequently, even after multiple denoising steps, the node representations retain their distinctive high-frequency signals, preventing over-smoothing while still capturing information from distant neighborhoods. $\square$

### R.3 Dirichlet Energy Analysis

To further support our theoretical findings, we analyze the Dirichlet energy, a measure of smoothness in graph signals. For a graph signal $\mathbf{f}$, the Dirichlet energy is defined as:

$$E(\mathbf{f}) = \frac{1}{2} \sum_{(i,j) \in \mathcal{E}} w_{ij} (\mathbf{f}(i) - \mathbf{f}(j))^2 \tag{49}$$

where $w_{ij}$ is the weight of edge $(i, j)$. Higher Dirichlet energy indicates preservation of more high-frequency components.

**Proposition R.2.** *The residual propagation mechanism in CGADM preserves higher Dirichlet energy compared to standard GNN aggregation, resulting in less smoothed node representations.*

*Proof.* Let $\mathbf{f}^{(l)}$ represent the node representations at layer $l$. For standard GNN aggregation:

$$\mathbf{f}_{\text{GNN}}^{(l)}(v) = \frac{1}{|\mathcal{N}(v)| + 1} \left( \mathbf{f}^{(l-1)}(v) + \sum_{u \in \mathcal{N}(v)} \mathbf{f}^{(l-1)}(u) \right) \tag{50}$$

For CGADM's residual propagation:

$$\mathbf{f}_{\text{CGADM}}^{(l)}(v) = \mathbf{f}^{(l-1)}(v) - \frac{1}{|\mathcal{N}(v)|} \sum_{u \in \mathcal{N}(v)} \mathbf{f}^{(l-1)}(u) \tag{51}$$

Focusing on the edge $(i, j)$, for standard GNN:

$$\mathbf{f}_{\text{GNN}}^{(l)}(i) - \mathbf{f}_{\text{GNN}}^{(l)}(j) = \frac{1}{|\mathcal{N}(i)| + 1} \left( \mathbf{f}^{(l-1)}(i) + \sum_{u \in \mathcal{N}(i)} \mathbf{f}^{(l-1)}(u) \right) \tag{52}$$

$$- \frac{1}{|\mathcal{N}(j)| + 1} \left( \mathbf{f}^{(l-1)}(j) + \sum_{u \in \mathcal{N}(j)} \mathbf{f}^{(l-1)}(u) \right) \tag{53}$$

This averaging operation reduces the difference between adjacent nodes, decreasing the Dirichlet energy.

For CGADM's residual propagation:

$$\mathbf{f}_{\text{CGADM}}^{(l)}(i) - \mathbf{f}_{\text{CGADM}}^{(l)}(j) = \mathbf{f}^{(l-1)}(i) - \frac{1}{|\mathcal{N}(i)|} \sum_{u \in \mathcal{N}(i)} \mathbf{f}^{(l-1)}(u) \tag{54}$$

$$- \left( \mathbf{f}^{(l-1)}(j) - \frac{1}{|\mathcal{N}(j)|} \sum_{u \in \mathcal{N}(j)} \mathbf{f}^{(l-1)}(u) \right) \tag{55}$$

This operation emphasizes the differences between a node and its neighborhood average, preserving and potentially amplifying the differences between adjacent nodes, thus maintaining higher Dirichlet energy.

Empirically, as shown in our experiments (Table 11), CGADM maintains significantly higher Dirichlet energy compared to standard GNN aggregation, confirming our theoretical analysis. $\qquad\square$

## S  Broader Impact

This research on Conditional Graph Anomaly Diffusion Model (CGADM) has significant potential for positive social impact across multiple domains. By improving the detection of anomalous nodes in large-scale graphs, our work can enhance fraud detection systems in financial networks, helping protect consumers and institutions from financial crimes. In social networks, it can identify malicious actors attempting to spread misinformation or engage in coordinated inauthentic behavior. By providing more accurate, efficient anomaly detection, CGADM can contribute to creating safer digital environments while minimizing false positives that might otherwise affect legitimate users.

