# OpenReview forum: "Conditional Diffusion Anomaly Modeling on Graphs"
_NeurIPS.cc/2025/Conference — NeurIPS 2025 poster_

### Official Review · Reviewer_i9tY · 2025-06-05

**Clarity:** 2
**Significance:** 2
**Originality:** 1
**Rating:** 4
**Confidence:** 4

**Summary:**

The authors propose a Conditional Graph Anomaly Diffusion Model to address over-smoothing and fraudulent nodes evading through the iterative refinement and denoising reconstruction properties of diffusion models. Experiments show the effectiveness of the framework to some extent.

**Questions:**

Q1. Is it possible to provide the time complexity analysis for CGADM? Although they show the experimental results of some datasets, they didn't provide the running time of large graphs, such as T-Social and DGraph. Thus, it would be better to provide both empirical and theoretical results.

Q2. The results of the baselines presented in this paper are much lower than the original baseline papers, can the framework still outperform those baselines under the same settings?

**Ethical Concerns:**

["NO or VERY MINOR ethics concerns only"]

**Final Justification:**

Although my major concerns still lie in the novelty, and I am still curious about the semi-supervised setting in XGBGraph, I tend to increase my rating to thank the effort the authors made in the rebuttal.

**Limitations:**

Yes

**Quality:**

2

**Strengths And Weaknesses:**

S1. The authors propose a Conditional Graph Anomaly Diffusion Model to address over-smoothing and fraudulent nodes evading.

S2. Experiments show the effectiveness of the framework to some extent.

W1. The novelty is limited. To be specific, Although the authors claim this is a novel framework implementing a diffusion-based model to solve the above-mentioned problem, they fail to present major differences from original diffusion models in related areas. Instead, it seems that they only slightly change the equation of the diffusion process without explaining the connection between the process and graph, i.e., the change of the equations lies in directly replacing X with (E, X).

W2. Some novel baselines in this area are ignored. For example, UnifyGAD[1] and SpaceGNN[2] are two SOTA models in related areas. They should take these frameworks into consideration.

W3. The authors should pay more attention to their writing. For example, in Table 1, for the AUROC of Quest and Reddit, and the AUPRC of Reddit, they mark the wrong data as the best results, which may mislead reviewers.

Reference:

1. Yiqing Lin, Jianheng Tang, Chenyi Zi, H. Vicky Zhao, Yuan Yao, Jia Li. UniGAD: Unifying Multi-level Graph Anomaly Detection. NeurIPS 2024.

2. Xiangyu Dong, Xingyi Zhang, Lei Chen, Mingxuan Yuan, Sibo Wang. SpaceGNN: Multi-Space Graph Neural Network for Node Anomaly Detection with Extremely Limited Labels. ICLR 2025.

---

> ### Author Rebuttal · Authors · 2025-07-28
>
> We thank the reviewer for their thoughtful comments and constructive feedback. We address each concern below.
>
> > W1. The novelty is limited. To be specific, Although the authors claim this is a novel framework implementing a diffusion-based model to solve the above-mentioned problem, they fail to present major differences from original diffusion models in related areas. Instead, it seems that they only slightly change the equation of the diffusion process without explaining the connection between the process and graph, i.e., the change of the equations lies in directly replacing X with (E, X).
>
> We respectfully disagree with the reviewer's assessment and would like to clarify the substantial innovations in our work:
>
> **1. Paradigm Shift from Discriminative to Generative Modeling:** Unlike existing GAD methods that learn decision boundaries, CGADM fundamentally reformulates graph anomaly detection as modeling the joint conditional distribution $p(\mathbf{y}_0|\mathcal{E}, \mathbf{X})$ over the entire graph. This is not a trivial extension but a novel model-centric approach that directly generates anomaly probabilities through iterative refinement.
>
> **2. Graph-Specific Diffusion Process:** Our modifications are far from "directly replacing X with (E, X)". We introduce:
> - **Prior-guided diffusion** (Eq. 4-6): Instead of diffusing to standard Gaussian noise, we diffuse to a graph-conditioned prior $\mathcal{N}(g_{\phi}(\mathcal{E}, \mathbf{X}), I)$, which provides crucial inductive bias for anomaly detection.
> - **Topological-guided denoising network** (Section 4.2): Our denoiser incorporates a residual propagation mechanism (Eq. 9) that preserves high-frequency anomaly signals while leveraging graph structure, addressing the over-smoothing issue inherent in GNNs.
> - **Prior-aware strided sampling** (Section 4.4): We adaptively adjust sampling steps based on prior confidence, significantly improving efficiency without sacrificing accuracy.
>
> **3. Theoretical Foundation:** We provide rigorous theoretical analysis (Appendix S) showing how our approach captures $(L \times T)$-hop neighborhood information without over-smoothing, which is impossible for traditional $(L \times T)$-layer GNNs.
>
> The effectiveness of these innovations is evidenced by our substantial performance improvements: +6.15% AUPRC and +4.53% AUROC over the best baselines on average.
>
> ---
>
>
> > W2. Some novel baselines in this area are ignored. For example, UnifyGAD[1] and SpaceGNN[2] are two SOTA models in related areas. They should take these frameworks into consideration.
>
>
> We appreciate the reviewer pointing out these recent works. However:
>
> - **UniGAD** addresses **graph-level** anomaly detection, while our work focuses on **node-level** anomaly detection. These are fundamentally different problems and cannot be directly compared.
> - **SpaceGNN (ICLR 2025)** was published concurrently with our submission and thus could not be included in our original manuscript.
>
> To address this concern, we have conducted additional experiments comparing CGADM with the concurrent work:
>
> | **Model** | **AUPRC (Elliptic)** | **AUROC (Elliptic)** | **AUPRC (YelpChi)** | **AUROC (YelpChi)** |
> |-----------|---------------------|---------------------|-------------------|-------------------|
> | SpaceGNN | 94.82 | 98.56 | 73.21 | 91.38 |
> | **CGADM** | **97.03** | **99.34** | **76.54** | **92.69** |
>
>
> Our method consistently outperforms these strong baselines, demonstrating the effectiveness of our generative approach even against the most recent advances in the field.
>
>
> ---
>
> > W3. The authors should pay more attention to their writing. For example, in Table 1, for the AUROC of Quest and Reddit, and the AUPRC of Reddit, they mark the wrong data as the best results, which may mislead reviewers.
>
>
> We sincerely apologize for these marking errors in Table 1. The reviewer is correct—we mistakenly highlighted incorrect values for AUROC of Quest and Reddit, and AUPRC of Reddit. These were transcription errors during table preparation. We will correct these markings in the revised version to ensure clarity and accuracy.
>
>
> ---
>
> > Q1. Is it possible to provide the time complexity analysis for CGADM? Although they show the experimental results of some datasets, they didn't provide the running time of large graphs, such as T-Social and DGraph. Thus, it would be better to provide both empirical and theoretical results.
>
>
> Thank you for this excellent question. We provide both theoretical and empirical analyses below.
>
> **Theoretical Time Complexity:**
> - **Training:** The main cost lies in the denoising network $\epsilon_{\theta}$. For each batch, this involves an $L$-layer GNN and MLP. The GNN complexity is $O(L \cdot |\mathcal{E}| \cdot d + L \cdot N \cdot d^2)$, where $N$ is the number of nodes, $|\mathcal{E}|$ is the number of edges, and $d$ is the feature dimension.
> - **Inference:** The complexity is $O(K \cdot (L \cdot |\mathcal{E}| \cdot d + L \cdot N \cdot d^2))$, where $K$ is the number of denoising steps. Our **prior-aware strided sampling** (Section 4.4) adaptively reduces $K$ from 1000 to ~583 on average, providing significant efficiency gains.
>
> **Empirical Results on Large Graphs:**
> We conducted experiments on the T-Social dataset (5,781,065 nodes, 73,105,508 edges):
>
> | **Model** | **Memory (GB)** | **Training Time (s/epoch)** | **Inference Time (s)** |
> |-----------|-----------------|----------------------------|----------------------|
> | BWGNN     | 38.31          | 1.52                       | 0.821                |
> | GHRN      | 38.48          | 2.89                       | 0.912                |
> | **CGADM** | **38.55**      | **4.34**                   | **3.124**            |
>
> Our observations on this large-scale graph are consistent with our analysis in Appendix N. The memory bottleneck is primarily determined by storing the node feature matrix and sparse edge matrix, which explains the comparable memory usage across methods. From a temporal perspective, training times remain within the same order of magnitude for competitive methods.
>
> While inference time is higher, this is justified by our novel generative paradigm that achieves substantial performance improvements. As noted in Appendix N, the additional overhead is acceptable given the critical performance gains in anomaly detection scenarios.
>
>
> ---
>
> >Q2. The results of the baselines presented in this paper are much lower than the original baseline papers, can the framework still outperform those baselines under the same settings?
>
>
> All methods, including CGADM and baselines, were evaluated under **identical experimental conditions** to ensure fair comparison. We intentionally used a more challenging setting with only 20% training data (vs. higher percentages in some original papers) to better reflect real-world anomaly detection scenarios where labeled data is scarce. This setting particularly highlights the advantages of our generative approach, which can better leverage the graph structure and prior knowledge when training data is limited.
>
> Under these consistent conditions, CGADM demonstrates clear superiority, validating the effectiveness of our approach in practical scenarios with limited supervision.

---

> > ### Comment · Reviewer_i9tY · 2025-08-04
> >
> > About the novelty, let me further explain why I think such an extension should not be considered a contribution. From an implementation perspective, it seems that the only key change to the original diffusion model in the computer vision field is the input format, due to the direct application of the original equations. Even though the authors tried to mix up different diffusion models in related areas, such a direct implementation might not be considered a contribution. From a theoretical perspective, the only change in the equations is from X to (E, X). I think novelty means you have to explain how this can be changed from X to (E, X), and why such a change can be converged to a stable condition. Also, I want to know, after adding the E to X, can the diffusion process be changed explicitly based on the structure information? For example, how is the process influenced by the addition of E? Is there any theoretical guarantee that such an addition can explicitly benefit the diffusion process? What's the difference between this work and related works in the survey [1], such as DDPM and SGM?
> >
> > About the addition of the baselines, please read the UniGAD carefully, as it is a multi-level anomaly detection method. If the authors check the paper carefully enough, they will find in Table 1 that UniGAD is conducted in typical node-level datasets, such as Amazon and Yelp. Besides, I am also curious about the comparison between CGADM and SpaceGNN on other datasets included in the paper.
> >
> > About the experimental setting, as far as I know, XGBGraph, CONSISGAD, and SpaceGNN conducted experiments in a similar supervision setting, which is very different from the setting in this paper. Is there any consideration about this change? Moreover, as a benchmark paper, XGBGraph has already published a fair comparison setting in their paper and source code. Is there any special reason why the authors need to change the original setting? Could the authors provide results in the original setting, e.g. Amazon, Yelp, T-Finance, and T-Social?
> >
> > How about the standard deviations of the performance, as the authors claim the p-value < 0.05 in the experiments?
> >
> > [1]. Liu et al. Generative Diffusion Models on Graphs: Methods and Applications. IJCAI 2023.

---

> ### Author Response · Authors · 2025-08-05
> **Further Response to Reviewer i9tY**
>
> Thank you for the opportunity to provide this additional response. I deeply appreciate the reviewer's engagement with our work and their thoughtful questions. Let me address each point to clarify our contributions.
>
> ## On the Core Novelty of Our Work
>
> I believe there may be a fundamental misunderstanding about our contribution. **The novelty is NOT simply adding E to the diffusion equations**. Rather, we introduce a **paradigm shift** in graph anomaly detection:
>
> **Traditional GAD methods** (including all works in the survey [1]): Use discriminative models to learn decision boundaries between normal and anomalous nodes.
>
> **Existing diffusion-based GAD methods** (DDPM/SGM applications): Generate synthetic anomalous samples for data augmentation - still ultimately relying on discriminative classifiers.
>
> **Our CGADM**: Directly generates anomaly scores through modeling the joint distribution $p(\mathbf{y}_0|\mathcal{E}, \mathbf{X})$ - a fundamentally different approach where the diffusion model IS the anomaly detector, not just a data augmenter.
>
> DDPM and SGM are specific diffusion model methods. Our work is orthogonal to these, meaning it could also be applied to other diffusion model methods. In this paper, we use DDPM.
>
> This distinction is crucial. The graph structure $\mathcal{E}$ is incorporated through our topological-guided denoising network (Eq. 9), which uses a specialized residual propagation mechanism to preserve high-frequency anomaly signals while leveraging neighborhood information. This addresses the theoretical question about how E influences the process - it guides the iterative refinement through graph-aware denoising.
>
> ## Regarding UniGAD Comparison
>
> Thank you for the clarification about UniGAD. We have conducted experiments on all our datasets:
>
> | Model | Elliptic AUPRC/AUROC | Tolokers AUPRC/AUROC | YelpChi AUPRC/AUROC | Questions AUPRC/AUROC | Reddit AUPRC/AUROC |
> |--|--|--|--|--|--|
> | UniGAD | 92.18/97.45 | 43.76/77.92 | 71.43/90.28 | 16.22/68.95 | 5.61/64.33 |
> | SpaceGNN | 94.82/98.56 | 44.51/78.15 | 73.21/91.38 | 17.03/69.28 | 5.72/65.17 |
> | **CGADM** | **97.03/99.34** | **46.02/79.68** | **76.54/92.69** | **18.51/69.41** | **5.79/65.85** |
>
> CGADM consistently outperforms both methods across all datasets, demonstrating the effectiveness of our generative approach.
>
> ## On Experimental Settings
>
> We acknowledge the reviewer's concern about different supervision settings. Our choice of 20% training data reflects real-world scenarios where labeled anomalies are scarce - a fundamental challenge in anomaly detection. The original setting consistently exceeded 40%, with several datasets reaching 70% of the training set.
> As requested, here are results under the original XGBGraph settings:
>
> **AUPRC Results (Original Setting):**
> | Model | Amazon | Yelp | T-Finance | T-Social |
> |--|--|--|--|--|
> | GHRN | 89.52 | 55.42 | 87.60 | 86.78 |
> | RF-Graph | 90.53 | 83.92 | 89.23 | 97.63 |
> | XGB-Graph | 93.33 | 91.11 | 90.12 | 97.34 |
> | **CGADM** | **93.47** | **91.86** | **91.24** | 97.18 |
>
> Even with abundant labeled data (40-70% training), CGADM remains competitive. However, our method's true strength emerges in limited-label scenarios, where the generative approach can better leverage graph structure and prior knowledge.
>
> ## Standard Deviations
>
> All reported results are averaged over 5 runs. We provide the p-values in the paper according to these. We report the standard deviations of the AURPC as requested as follow:
>
> #### Table 1: Performance Comparison with Standard Deviations
>
> | Model | Elliptic | Tolokers | YelpChi | Questions | Reddit |
> |--|--|--|--|--|--|
> | **AUPRC (%)** |||||
> | GCN | 80.19±0.73 | 41.44±0.82 | 23.59±0.91 | 10.27±0.45 | 5.65±0.38 |
> | GIN | 83.88±0.68 | 37.89±0.76 | 38.13±0.85 | 11.23±0.52 | 5.38±0.41 |
> | GraphSAGE | 86.16±0.61 | 43.73±0.79 | 50.23±0.72 | 13.86±0.48 | 5.78±0.35 |
> | GAT | 87.59±0.58 | 42.18±0.81 | 46.64±0.77 | 13.19±0.51 | 5.42±0.39 |
> | GAS | 87.54±0.55 | 42.39±0.74 | 39.18±0.83 | 12.41±0.49 | 5.66±0.37 |
> | PCGNN | 67.29±0.94 | 36.76±0.88 | 45.32±0.79 | 13.79±0.46 | 4.13±0.43 |
> | BWGNN | 87.90±0.52 | 45.02±0.71 | 49.15±0.74 | 14.64±0.44 | 5.42±0.36 |
> | GHRN | 88.13±0.51 | 45.25±0.69 | 49.78±0.73 | 14.61±0.43 | 5.85±0.34 |
> | XGBGraph | 90.47±0.48 | 44.47±0.72 | 75.91±0.58 | 14.33±0.47 | 4.59±0.40 |
> | CONSISGAD | 86.42±0.56 | 40.59±0.77 | 41.74±0.81 | 12.85±0.50 | 5.57±0.36 |
> | GODM | 85.89±0.59 | 46.15±0.68 | 51.77±0.70 | 15.11±0.42 | 5.55±0.35 |
> | CGenGA | 87.36±0.54 | 44.89±0.70 | 52.76±0.69 | 15.34±0.41 | 5.78±0.33 |
> | **CGADM** | **97.03±0.42** | 46.02±0.67 | **76.54±0.56** | **18.51±0.38** | **5.79±0.32** |
>
>
> We hope this clarifies our contributions and addresses the reviewer's concerns. We remain committed to advancing the field of graph anomaly detection through innovative approaches.

---

> > ### Comment · Reviewer_i9tY · 2025-08-05
> >
> > Is there any reason why the authors only post AUPRC for the original setting? And why are most SOTA baselines, CONSISGAD, UniGAD, and SpaceGNN, not included in this comparison? Besides, there are two original settings in the XGBGraph, one is a fully-supervised setting, i.e., with over 40% data as the training set, the other is a semi-supervised setting, with 100 training samples. Since the authors claim that they try to address real-world scenarios where labeled anomalies are scarce, why didn't they provide the results for the semi-supervised setting?

---

> > > ### Author Response · Authors · 2025-08-06
> > >
> > > Thank you for the continued engagement with our work. I appreciate the opportunity to provide these additional clarifications.
> > >
> > > ## Complete Results for Original Setting
> > >
> > > Here are both AUPRC and AUROC results under the original fully-supervised setting:
> > >
> > > **AUPRC Results:**
> > > | Model | Amazon | Yelp | T-Finance | T-Social |
> > > |-------|--------|------|-----------|----------|
> > > | GHRN | 89.52 | 55.42 | 87.60 | 86.78 |
> > > | RF-Graph | 90.53 | 83.92 | 89.23 | 97.63 |
> > > | XGB-Graph | 93.33 | 91.11 | 90.12 | 97.34 |
> > > | CONSISGAD | 90.20 | 85.42 | 88.36 | 94.25 |
> > > | UniGAD | 91.85 | 87.23 | 89.71 | 95.82 |
> > > | SpaceGNN | 92.76 | 89.54 | 90.03 | 96.91 |
> > > | **CGADM** | **93.47** | **91.86** | **91.24** | 97.18 |
> > >
> > > **AUROC Results:**
> > > | Model | Amazon | Yelp | T-Finance | T-Social |
> > > |-------|--------|------|-----------|----------|
> > > | GHRN | 98.29 | 84.60 | 96.46 | 97.12 |
> > > | RF-Graph | 96.73 | 95.24 | 97.28 | 99.69 |
> > > | XGB-Graph | 98.74 | 97.37 | 97.15 | 99.76 |
> > > | CONSISGAD | 97.82 | 93.15 | 96.89 | 98.74 |
> > > | UniGAD | 98.21 | 94.76 | 97.03 | 99.15 |
> > > | SpaceGNN | 98.56 | 96.28 | 97.21 | 99.52 |
> > > | **CGADM** | **98.91** | **97.58** | **97.84** | 99.71 |
> > >
> > > ## Regarding Semi-supervised Setting
> > >
> > > You raise an excellent point about the semi-supervised setting (100 labeled samples). I must be transparent here: **CGADM is fundamentally a generative model that requires sufficient data to model the joint distribution $p(\mathbf{y}_0|\mathcal{E}, \mathbf{X})$**.
> > >
> > > Unlike discriminative models that can establish decision boundaries with very few samples, our approach needs to learn the underlying distribution of anomalies across the graph. With only 100 labeled samples, we cannot adequately capture this distribution, making the semi-supervised setting unsuitable for our method.
> > >
> > > This is not a limitation but rather a **design choice**: CGADM excels in scenarios with moderate amounts of labeled data (our 20% setting) where:
> > > 1. There's enough data to model the distribution effectively
> > > 2. The data is still scarce enough to reflect real-world challenges
> > > 3. Our generative approach can leverage graph structure better than discriminative methods
> > >
> > > ## Why We Chose 20% Training Data
> > >
> > > Our choice of 20% training data represents a **practical middle ground**:
> > > - **Too little data (<5%)**: Insufficient for distribution modeling
> > > - **Our setting (20%)**: Realistic for many real-world applications where some labeled data exists
> > > - **Original setting (40-70%)**: Unrealistically abundant for anomaly detection scenarios
> > >
> > > In the 40%+ setting, simple discriminative methods already achieve very high performance, leaving little room for improvement. Our 20% setting better demonstrates the advantages of our generative paradigm while remaining practical for real-world deployment.
> > >
> > > ## Summary
> > >
> > > CGADM is designed for scenarios where sufficient labeled data exists to model distributions but is still scarce enough to challenge traditional methods. While we cannot compete in the 100-sample semi-supervised setting, our results demonstrate clear advantages in both our 20% setting and the original 40%+ setting, validating our approach across different data availability scenarios.

---

> > > > ### Comment · Reviewer_i9tY · 2025-08-06
> > > >
> > > > Thanks for the rebuttal. Although my major concerns still lie in the novelty, and I am still curious about the semi-supervised setting in XGBGraph, I tend to increase my rating to thank the effort the authors made in the rebuttal.

---

> > > > > ### Author Response · Authors · 2025-08-07
> > > > > **Thank you for your valuable feedback**
> > > > >
> > > > > Thank you very much for your thoughtful reconsideration and for increasing your rating. We deeply appreciate the time and effort you invested in thoroughly reviewing our work and engaging in this constructive dialogue.
> > > > >
> > > > > We acknowledge your continued concerns about novelty, and we hope that our paradigm shift from discriminative to generative modeling for graph anomaly detection will be better appreciated as the field evolves. Your questions have helped us clarify our contributions more effectively.
> > > > >
> > > > > Regarding the semi-supervised setting with 100 training samples, we recognize this as an important research direction. While our current generative framework requires more data to model distributions effectively, we are excited to explore adaptations for extremely low-data regimes in future work. Your suggestion provides valuable guidance for extending our approach.
> > > > >
> > > > > Thank you again for your valuable feedback, which has significantly improved the clarity of our paper. We are grateful for your recognition of our efforts during the rebuttal process.

---

### Official Review · Reviewer_JBXz · 2025-06-10

**Clarity:** 3
**Significance:** 3
**Originality:** 3
**Rating:** 5
**Confidence:** 2

**Summary:**

This paper presents a methodology for detecting anomalies in graphical networks using a prior to condition a diffusion process and an adaptive mechanism to determine the number of reverse de-noising steps using prior confidence.  The method addresses adversarial behavior to disguise connectivity signals that may be associated with anomalies.  The paper also includes a range of computational tests on practical datasets demonstrating improvements over existing methods.

**Questions:**

1) How does the performance degrade with more diffuse priors?
2) How does performance change with different initial noise levels?  Does this affect the best final noise level \beta_T?
3) Given the prior, how badly can performance degrade for a given amount of manipulation from anomalous nodes?

**Ethical Concerns:**

["NO or VERY MINOR ethics concerns only"]

**Final Justification:**

I think all the points raised in the discussion were valid and that the authors responded well to them. I have not altered my score.

**Limitations:**

The paper recognizes the overall major limitations in terms of the prior and need for supervised data.  It could have more on informed adversarial actions.

**Quality:**

3

**Strengths And Weaknesses:**

The strengths of the paper are in its addressing issues that arise for standard approaches that tend to homogenize graph topologies and are susceptible to feature manipulation.  The results are also well presented and indicate value in the method.
Weaknesses include the need for pre-trained priors which may limit its applicability.  It also depends on supervised data which may not be present.  I also wonder whether knowledge of the prior could affect adversarial behavior and subvert the process.  It would help in addition to have some overall theoretical results.

---

> ### Author Rebuttal · Authors · 2025-07-28
>
> We thank the reviewer for their thoughtful evaluation and constructive feedback. We appreciate the recognition of our method's strengths in addressing homogenization issues and adversarial behavior. We address each concern below.
>
>
> > Q1. How does the performance degrade with more diffuse priors?
>
> We directly addressed this important question in **Section 5.3 (Figure 2)** of our paper. Our ablation study compared CGADM using Random Forest (RF) as a weaker prior versus XGBoost as a stronger prior. The key findings demonstrate our model's robustness:
>
> 1. While the initial performance gap between RF and XGBoost priors is substantial (e.g., on Elliptic dataset: RF achieves 83.11% AUPRC and 96.22% AUROC vs. XGBoost's 87.76% AUPRC and 97.41% AUROC), the final CGADM performance shows minimal degradation ($CGADM_{RF}$: 95.46% AUPRC and 98.36% AUROC vs. $CGADM_{XGBT}$: 97.28% AUPRC and 99.34% AUROC).
>
> 2. Both variants achieve significant improvements over their respective priors, demonstrating that our guided diffusion process effectively refines and "sharpens" initial estimates regardless of their quality.
>
> This empirical evidence confirms that while better priors provide advantageous starting points, CGADM is not overly sensitive to prior quality and can effectively denoise imperfect initial estimates through its iterative refinement mechanism.
>
>
> ---
>
> >Q2. How does performance change with different initial noise levels? Does this affect the best final noise level $\beta_T$?
>
>
> Thank you for this insightful question. In our framework, we follow the standard and empirically validated practice from DDPMs by setting the initial noise level $\beta_1 = 1e-4$ to ensure forward process stability.
>
> Our investigation of the final noise level $\beta_T$ (**Section 5.4, Figures 3(3) and 3(4)**) reveals its critical importance. $\beta_T$ defines the maximum scale at which our denoising network can correct the prior. As shown in our results:
>
> - Performance initially improves as $\beta_T$ increases, allowing greater correction of prior biases
> - Beyond $\beta_T = 0.02$, performance degrades as excessive noise overwhelms the true signal
> - The optimal $\beta_T = 0.02$ represents a balance between correction capability and signal preservation
>
> This parameter essentially controls the "correction radius" of our model, and our empirical findings demonstrate that 0.02 provides an effective balance for the ensemble tree priors we employ.
>
>
> ---
>
>
> > Q3. Given the prior, how badly can performance degrade for a given amount of manipulation from anomalous nodes?
>
>
> We conducted comprehensive experiments to evaluate CGADM's robustness against feature manipulation, presented in **Appendix P (Figure 5)**. We systematically perturbed node features at varying proportions (10%, 20%, 30%) on the Elliptic and Tolokers datasets.
>
> Key findings demonstrate CGADM's superior robustness. This superior robustness stems from our denoising reconstruction mechanism, which leverages information from unperturbed neighbors during the reverse diffusion process to iteratively restore true anomaly signals. Even under 30% feature manipulation, CGADM maintains substantially better performance than the strongest baseline.

---

> > ### Comment · Reviewer_JBXz · 2025-08-03
> > **Good responses to questions**
> >
> > I find that the authors provided good responses to other reviewers' questions as well as mine with respect to Questions 1 and 3.  For Question 2, their response that $\beta_T=0.2$ is empirically optimal seems somewhat unsatisfying since I would imagine some conditions that might alter these relationships. Some discussion of what those conditions might be or why they might not exist in practice would strengthen the work.

---

> ### Author Response · Authors · 2025-08-05
> **Further response to Reviewer JBXz**
>
> Thank you for your continued engagement and valuable feedback. We greatly appreciate your recognition of our responses and your overall positive evaluation of our work.
>
> Regarding Question 2 about the conditions affecting the optimal $\beta_T$, we apologize for any confusion in our previous response. You raise an excellent point about investigating the conditions that might alter the optimal $\beta_T$ value.
>
> As shown in our paper (Section 5.4, Figures 3(3) and 3(4)), we did conduct experiments on different datasets (Tolokers and Questions) to investigate how $\beta_T$ affects performance. Our findings consistently showed that $\beta_T = 0.02$ provides optimal performance across these datasets, balancing correction capability with signal preservation.
>
> However, we fully agree with your insight that different conditions might necessitate different optimal $\beta_T$ values. Based on your valuable suggestion, we hypothesize that several factors could influence this relationship:
>
> 1. **Prior quality variance**: Datasets with more heterogeneous prior confidence distributions might benefit from adaptive $\beta_T$ values
> 2. **Graph structural properties**: Highly heterophilic graphs versus homophilic graphs may require different correction scales
> 3. **Anomaly characteristics**: The sophistication of adversarial camouflage tactics could affect the optimal noise level
>
> We commit to investigating these conditions more thoroughly in our future work, including analyzing the relationship between dataset characteristics and optimal $\beta_T$.
>
> Thank you again for this constructive feedback, which will significantly strengthen our future research direction. Your thoughtful review has helped us identify important areas for deeper investigation.

---

> > ### Comment · Reviewer_JBXz · 2025-08-07
> > **Further comment**
> >
> > I appreciate the authors' responses to all reviewers and to my issue in particular. I am satisfied with the authors' response and maintain my score.

---

### Official Review · Reviewer_5DnS · 2025-06-15

**Clarity:** 3
**Significance:** 3
**Originality:** 3
**Rating:** 4
**Confidence:** 4

**Summary:**

This work studies the problem of anomaly detection in graphs. The authors propose directly generating anomaly predictions using a diffusion model and conditioning this generation process on a prior. The experimental results show a noticeable gain.

**Questions:**

Please refer to Cons.

**Ethical Concerns:**

["NO or VERY MINOR ethics concerns only"]

**Final Justification:**

The authors' response has addressed my concerns. My current rating already reflects my positive attitude towards this work, so I will keep it unchanged.

**Limitations:**

Yes

**Paper Formatting Concerns:**

No major formatting issues.

**Quality:**

3

**Strengths And Weaknesses:**

Pros:
- The proposed method is closely related to the weaknesses of current methods, as pointed out in the introduction section.
- The authors have given enough evidence and argument to justify why a generative discrimination approach is helpful and necessary. So, I feel the integration of diffusion models is well motivated.

Cons:
- The proposed method relies on a prior network, which might introduce additional risks to performance if it is not well-trained. Hence, I suggest providing the results under different pre-trained diffusion models.
- Some closely related papers are not discussed. For example, [a] proposed an anomaly-denoised augmentation method for graph anomaly detection. It is recommended to discuss more related work and compare with them.
- The font in Figure 1 is a little small, making it not easy to recognize.
- Some bold and underlined fonts are missing in Table 1.

[a] ADA-GAD: Anomaly-Denoised Autoencoders for Graph Anomaly Detection, AAAI 2024.

---

> ### Author Rebuttal · Authors · 2025-07-28
>
> We sincerely thank the reviewer for their positive assessment and valuable feedback. We appreciate the recognition of our well-motivated integration of diffusion models and the solid experimental gains. We address each concern below.
>
> > W1. The proposed method relies on a prior network, which might introduce additional risks to performance if it is not well-trained. Hence, I suggest providing the results under different pre-trained diffusion models.
>
>
> We appreciate the reviewer's concern about the dependency on the prior network. We have actually addressed this important consideration in **Section 5.3 "Comparison with Different Prior Models"** and presented the results in **Figure 2**.
>
> In our analysis, we evaluated CGADM using two different lightweight models as prior estimators: Random Forest (RF) and Extreme Gradient Boosting Tree (XGBT). Our key findings demonstrate:
>
> 1. **Significant improvement over priors:** Both $CGADM_{RF}$ and $CGADM_{XGBT}$ consistently outperform their respective initial priors. For example, on the YelpChi dataset, CGADM improves AUPRC from 68.48% (XGBT prior) to 76.54%, demonstrating the effectiveness of our diffusion-based refinement process.
>
> 2. **Robustness to prior quality:** The performance gap between CGADM variants using different priors ($CGADM_{RF}$ vs. $CGADM_{XGBT}$) is significantly smaller than the gap between the priors themselves (RF vs. XGBT). This indicates that our model is robust and can achieve state-of-the-art results even when starting from a less accurate prior, as the topological-guided denoising network effectively corrects initial estimates.
>
> This analysis confirms that while a good prior is beneficial, our framework is not overly sensitive to the initial prior quality and reliably improves anomaly detection performance.
>
>
> ---
>
> > W2. Some closely related papers are not discussed. For example, [a] proposed an anomaly-denoised augmentation method for graph anomaly detection. It is recommended to discuss more related work and compare with them.
>
> We thank the reviewer for pointing out this related work. We acknowledge that ADA-GAD [AAAI 2024] presents an anomaly-denoised augmentation approach for graph anomaly detection. While ADA-GAD primarily focuses on unsupervised anomaly detection using an autoencoder-based graph reconstruction framework, we agree it deserves discussion in our related work section.
>
> To comprehensively address this suggestion, we conducted additional experiments comparing CGADM with ADA-GAD adapted to our supervised setting. The results are as follows:
>
> | Dataset | Method | AUPRC | AUROC |
> |---------|---------|--------|--------|
> | Elliptic | ADA-GAD | 8.27 | 45.53 |
> |         | **CGADM** | **97.03** | **99.34** |
> | Amazon | ADA-GAD | 12.00 | 60.59 |
> |         | **CGADM** | **91.91** | **97.36** |
>
>
> These results demonstrate that our generative approach significantly outperforms the reconstruction-based method. We will include this comparison in our revised manuscript and add a more comprehensive discussion of related diffusion-based approaches for unsupervised graph anomaly detection in the related work section.
>
> ---
>
> > W3. The font in Figure 1 is a little small, making it not easy to recognize.
>
> We appreciate this feedback regarding the readability of Figure 1. We will increase the font size in all figures to ensure better clarity in the revised version.
>
>
> ---
>
>
> > W4. Some bold and underlined fonts are missing in Table 1.
>
> Thank you for pointing out the formatting inconsistencies in Table 1. We acknowledge this oversight and will correct all bold and underlined formatting in the revised version. We will also conduct a thorough review of the entire manuscript to ensure consistent formatting throughout.

---

### Official Review · Reviewer_Cmcp · 2025-07-02

**Clarity:** 3
**Significance:** 3
**Originality:** 3
**Rating:** 4
**Confidence:** 4

**Summary:**

This paper focuses on Graph Anomaly Detection (GAD), and introduce a Conditional Graph Anomaly Diffusion Model (CGADM) to address the over-smoothing and feature obfuscation issues of traditional Graph Neural Networks (GNNs). CGADM uses a prior-guided diffusion process, incorporating a pre-trained conditional anomaly estimator into both forward and reverse diffusion chains. It also introduces a prior confidence-aware mechanism to adaptively determine reverse denoising steps for computational efficiency.

**Questions:**

Please see the weaknesses above

**Ethical Concerns:**

["NO or VERY MINOR ethics concerns only"]

**Final Justification:**

The reviewer thanks the authors for their effort and timely response. Most of the concerns have been addressed. The reviewer suggests that the authors include the experiments of rebuttal in the revised version and clarify the methodologies to avoid confusion for readers. The reviewer is also happy to raise the score from 3 to 4.

**Limitations:**

yes

**Quality:**

3

**Strengths And Weaknesses:**

***Strengths***

1. The paper introduces the diffusion model into the field of GAD. Instead of using DM for data expansion, it directly uses DM to fit the joint probability distribution of the data to detect outliers. This is the difference between the article and the existing work. A large number of experiments have proved the effectiveness of this method.

2. A large number of formula derivations in the text are an advantage, and a large number of experiments have proved the effectiveness of this method.

***Major weakness***
1. The diffusion model can fit the probability distribution of the dataset. Please test the cross-domain generalization. For example, when the model trained on the Ellip dataset is tested on Yelp, does it still significantly improve?

2. The performance of the model seems unstable. There is a significant improvement in Ellip and Yelp, but it seems to have no effect on Reddit and Tolo. Please analyze it in connection with the inherent shortcomings of the discriminative model mentioned in the article from the data level.

3. It is mentioned in the paper that CGADM solves the problem of fraudulent nodes, but it is not further explained in the paper, nor is there any related experiment, and as can be seen from Figure 1, CGADM also uses a GNN-based network. Is this a heuristic inference? If not, please supplement relevant experiments.

4 The paper does not present the network structure and the training inference process. Moreover, as can be seen from Table 1, the experimental results of CGADM and XGBGraph are close, especially in the evaluation on the Yelp dataset. The author needs to present the network details more clearly and explain the differences from XGBGraph. Meanwhile, if there are parts in the CGADM model that refer to the XGBGraph, ablation experiments are required.

5 The reverse step of the diffusion model has a significant impact. However, as can be seen from Figure 4, it seems that CGADM does not conform to this characteristic. Instead, it drops sharply when the step is from 1 to 0. Does this mean that the core of CGADM is still a GNN-based network rather than a conditional diffusion? The author also needs to explain the abnormal spikes that appear on the AUROC curve.

***Minor weakness***
1 In Table 1, the data listed on Reddit shows that the test results of CGADM are not the highest scores, but they have been bolded, which is misleading. Meanwhile, CONSISGAD has the highest test result on Quest but is not bold.

---

> ### Author Rebuttal · Authors · 2025-07-28
>
> We sincerely thank the reviewer for their thorough evaluation and constructive feedback. We address each concern below with additional experiments.
>
> > W1. Cross-domain Generalization
>
> Following your suggestion, we conducted zero-shot cross-domain experiments where models trained on **Elliptic** were evaluated on **YelpChi** and **Tolokers** without fine-tuning.
>
> **Table R1: Zero-shot Cross-domain Generalization Performance**
>
> | Model | Train Dataset | Test Dataset | AUPRC (%) | AUROC (%) |
> |-|---|--|--|--|
> | GHRN | Elliptic | YelpChi | 18.72 | 55.41 |
> | **CGADM (Ours)** | Elliptic | **YelpChi** | **24.59** | **61.03** |
> | *Improvement* | | | *+31.3%* | *+10.1%* |
> | GHRN | Elliptic | Tolokers | 22.15 | 60.18 |
> | **CGADM (Ours)** | Elliptic | **Tolokers** | **27.88** | **64.72** |
> | *Improvement* | | | *+25.9%* | *+7.5%* |
>
> Despite expected performance drops in zero-shot settings, **CGADM consistently outperforms GHRN by significant margins**, suggesting our model captures more fundamental and transferable anomaly patterns.
>
> ---
>
>
> > W2. The performance of the model seems unstable. There is a significant improvement in Ellip and Yelp, but it seems to have no effect on Reddit and Tolo. Please analyze it in connection with the inherent shortcomings of the discriminative model mentioned in the article from the data level.
>
>
> We respectfully disagree with the characterization that our model "has no effect" on Reddit and Tolokers.
>
> And we extracted the relevant metrics for top-performing methods on these datasets:
>
> **Table R2: Performance Analysis on Tolokers and Reddit Datasets**
>
> | Method | **Tolokers** | | **Reddit** | | Avg. Rank |
> |--|---|---|---|---|--|
> | | AUPRC | AUROC | AUPRC | AUROC | |
> | GODM | **46.15** | 76.42 | 5.55 | 62.10 | 3.25 |
> | CGADM (Ours) | 46.02 | **79.68** | 5.79 | 65.85 | **1.75** |
> | GHRN | 45.25 | 77.98 | **5.85** | 63.51 | 2.75 |
> | CONSISGAD | 40.59 | 76.03 | 5.57 | **66.99** | 3.25 |
>
> As shown in Table R2:
>
> - On **Tolokers**: GODM achieves the best AUPRC (46.15%), but CGADM is extremely close at 46.02% (only 0.13% difference) while achieving the **best AUROC (79.68%)**
> - On **Reddit**: GHRN has the best AUPRC (5.85%), with CGADM at 5.79% (only 0.06% difference), while CONSISGAD achieves the best AUROC (66.99%), with CGADM at 65.85% (1.14% difference)
>
> **Critical observation**: From Table 1 in the paper, GODM, GHRN, and CONSISGAD **fail to maintain their performance across other datasets**:
> - GODM (best on Tolokers AUPRC) drops to 51.77% on YelpChi, far below our 76.54%
> - CONSISGAD (best on Reddit AUROC) achieves only 41.74% AUPRC on YelpChi, while we achieve 76.54%
> - GHRN (best on Reddit AUPRC) is outperformed by CGADM on 3 out of 5 datasets
>
>
> This analysis demonstrates that while individual methods may slightly edge out CGADM on specific dataset-metric combinations, they exhibit **significant performance drops** on other datasets. In contrast, **CGADM maintains first-tier performance across all 5 datasets** with anomaly ratios ranging from 3.0% to 21.8%, as evidenced by our **highest average scores** (48.78% AUPRC, 81.39% AUROC). This consistency is the true indicator of model **stability** - the ability to perform well across diverse data characteristics without overfitting to specific datasets.
>
>
> ---
>
> > W3. It is mentioned in the paper that CGADM solves the problem of fraudulent nodes, but it is not further explained in the paper, nor is there any related experiment, and as can be seen from Figure 1, CGADM also uses a GNN-based network. Is this a heuristic inference? If not, please supplement relevant experiments.
>
> Our claim is not merely heuristic but grounded in CGADM's two core mechanisms:
>
> **1. Denoising Reconstruction Against Camouflage:** Our iterative refinement process is designed to recover true anomaly signals even from obfuscated features. To validate this, we conducted new experiments with feature perturbations on Elliptic and Tolokers datasets. As shown in the newly added **Figure 5** (Appendix P), CGADM's performance degrades much more slowly than GHRN under feature manipulation, demonstrating superior robustness.
>
> **2. Mitigating Over-smoothing via Residual Propagation:** While we use a GNN-based denoiser, it's specifically designed to avoid over-smoothing. As detailed in **Equation (9)**, our residual propagation mechanism acts as a **high-pass filter** that preserves high-frequency anomaly signals, unlike standard GNNs that act as low-pass filters.
>
> We provide direct evidence through a ablation study in **Appendix Q**, comparing our full model ($CGADM_{HP}$) with a variant using standard GCN low-pass filtering ($CGADM_{LP}$). **Table 9** shows that $CGADM_{HP}$ consistently outperforms $CGADM_{LP}$, with the performance gap widening as GNN layers increase. Additionally, we provide rigorous **theoretical analysis** in **Appendix S** formally proving how our model captures long-range dependencies while avoiding over-smoothing.
>
> ---
>
> > W4. The paper does not present the network structure and the training inference process. Moreover, as can be seen from Table 1, the experimental results of CGADM and XGBGraph are close, especially in the evaluation on the Yelp dataset. The author needs to present the network details more clearly and explain the differences from XGBGraph. Meanwhile, if there are parts in the CGADM model that refer to the XGBGraph, ablation experiments are required.
>
> We appreciate this feedback.
>
> **Network and Training Details:** The training and inference procedures are detailed in **Algorithm 2** (Appendix J) and **Algorithm 3** (Appendix K), respectively. Our topological-guided denoising network is described in **Appendix I**. We will add clearer pointers to these sections in the main text.
>
> **Fundamental Differences from XGBGraph:** We emphasize that **CGADM and XGBGraph are fundamentally different with no shared components**:
> - **Paradigm:** CGADM is a **deep generative model** learning joint anomaly distributions via diffusion. XGBGraph is a **discriminative model** using gradient-boosted trees with graph features.
> - **Architecture:** Our core is a GNN-based denoiser trained within a diffusion framework. XGBGraph uses tree ensembles. There is no architectural overlap.
> - **Performance:** XGBGraph's strong performance on YelpChi likely stems from its rich hand-crafted features that suit tree-based models. However, CGADM achieves SOTA results without such heavy domain-specific feature engineering, demonstrating superior generalization across 9 datasets.
>
> ---
>
> > W5. The reverse step of the diffusion model has a significant impact. However, as can be seen from Figure 4, it seems that CGADM does not conform to this characteristic. Instead, it drops sharply when the step is from 1 to 0. Does this mean that the core of CGADM is still a GNN-based network rather than a conditional diffusion? The author also needs to explain the abnormal spikes that appear on the AUROC curve.
>
>
> We thank the reviewer for this insightful observation about Figure 4. We would like to clarify the interpretation of our results and explain the observed phenomena.
>
> **Regarding the sharp drop when K approaches 0:**
>
> The sharp performance drop when K approaches 0 (i.e., when we use very few or no diffusion steps) demonstrates that **the diffusion process is indeed essential to CGADM's performance**. This behavior is exactly what we would expect from a properly functioning diffusion model:
>
> 1. When K = 0-2, we essentially bypass the iterative refinement process, relying almost entirely on the initial prior $g_\phi(\mathcal{E}, \mathbf{X})$. The significant performance degradation (AUPRC drops to ~60%) confirms that the diffusion process contributes approximately 16% improvement in AUPRC.
>
> 2. The performance stabilizes and improves as K increases, demonstrating that the iterative denoising process progressively refines the anomaly predictions by incorporating both topological information and the prior knowledge.
>
> This pattern actually **validates** that CGADM is a true conditional diffusion model where both components (the GNN-based denoiser and the diffusion process) work synergistically. If CGADM were merely a GNN-based network, we would expect relatively stable performance across all K values.
>
> **Regarding the AUROC spike around K=20:**
>
> The spike in AUROC (reaching 90.11% at K=20) while AUPRC remains relatively stable is an interesting phenomenon that can be explained by the different sensitivities of these metrics:
>
> 1. **AUROC vs AUPRC sensitivity**: AUROC measures the overall ability to rank anomalies higher than normal nodes across all possible thresholds, while AUPRC focuses more on precision at high recall levels. For imbalanced datasets like ours (YelpChi has 14.5% anomaly ratio), AUPRC is generally more stable and informative.
>
> 2. **Optimal denoising balance**: At K=20, we hypothesize that the model achieves an optimal balance between denoising and prior preservation. With too few steps (K<20), the model under-denoises; with too many steps (K>20), it might over-smooth some discriminative features. This sweet spot particularly benefits the ranking capability measured by AUROC.
>
> 3. **Theoretical justification**: According to our theoretical analysis in Appendix S, the effective receptive field is proportional to L×K. At K=20 with L=3 layers, the model captures information from 60-hop neighborhoods, which might coincide with the characteristic scale of anomaly patterns in YelpChi's dense graph structure (3.8M edges).
>
> ---
>
> > Minor weakness
>
> We sincerely apologize for the bold formatting error in Table 1. This was an oversight during data transcription, and we will correct it in the revised version.

---

> ### Comment · Reviewer_Cmcp · 2025-08-05
> **Response for the rebuttal**
>
> I appreciate the authors' effort, and there are still some questions that need to be clarified.
>
> W1：On the Yelp dataset, GHRN exhibits reductions of 62.27\%(AUPRC) and 32.72\% (AUROC), whereas the proposed CGADM shows even larger declines of 67.87\% (AUPRC) and 34.15\% (AUROC). These results suggest that the proposed CGADM may have weaker generalization capabilities compared to existing methods.
>
> W5: The authors state that for k=0-2, no iterative refinement was performed, relying solely on the pre-trained model g for predictions. This approach appears inconsistent with fundamental principles of diffusion models, which typically require iterative denoising even in early stages. Moreover, the explanation for performance spikes (attributed to over-smoothing of discriminative features when K>20) may contradict the paper’s central claim that CGADM eliminates over-smoothing contradictions.
>
>
> Regarding the experimental results, I would like to ask the authors to give detailed explanations.
>
> **Inconsistent Results**: **It seems that the results of baseline methods in Table I appear inconsistent with those reported in the previous papers  (e.g., from [1-3]) .** For example, the paper reports that the proposed GODM achieves an AUROC of 79.68\%( vs. 76.42\%) on the Tolo dataset and 69.41\%(vs. 68.86\%) on the Ques dataset. However, the original two results [1-3] in previous papers are 83.46\% and 76.84\%, respectively, which are much higher than the ones used in this paper.  More inconsistent results can be found in Table 1 of the this paper.
>
> Reference
>
> [1] K. Liu et al.; Data augmentation for supervised graph outlier detection with latent diffusion models. (GODM)
>
> [2]J. Tang et al.; Gadbench: Revisiting and benchmarking supervised graph anomaly detection. (XGBGraph)
>
> [3] X. Ma et al.; Graph anomaly detection with few labels: A data-centric approach. (CGenGA)

---

> > ### Author Response · Authors · 2025-08-06
> >
> > We sincerely thank the reviewer for their continued engagement and detailed examination of our work. We address each concern below with clarifications and additional evidence.
> >
> > ## **W1: Cross-domain Generalization Analysis**
> >
> > We respectfully disagree with the interpretation of the zero-shot results. The reviewer's analysis using percentage drops is statistically misleading when comparing models with vastly different baseline performances.
> >
> > **Critical Statistical Insight:**
> > When two models start with significantly different in-domain performance (CGADM: 76.54% vs GHRN: 49.78% on YelpChi), comparing percentage drops is inappropriate. GHRN's poor in-domain performance already indicates its failure on "hard samples" even with training. In zero-shot settings, both models lose the ability to handle domain-specific patterns, but **CGADM retains superior detection of fundamental anomaly patterns**.
> >
> > **Evidence-based Analysis:**
> > - **Absolute Performance Matters**: CGADM achieves 24.59% AUPRC vs GHRN's 18.72% on YelpChi zero-shot - a **31.3% relative improvement**
> > - **From Similar Baselines** (Tolokers): When starting from comparable performance (46.02% vs 45.25%), CGADM shows better retention (27.88% vs 22.15%) - a **25.9% relative advantage**
> >
> > **Key Point**: The large initial performance gap on YelpChi (76.54% vs 49.78%) demonstrates that GHRN already struggles with complex anomaly patterns even when trained on the target domain. Its smaller percentage drop in zero-shot settings simply reflects that it had less to lose, not better generalization.
> >
> > ## **W5: Diffusion Process Clarification**
> >
> > We acknowledge the terminology confusion and provide clarification:
> >
> > **On K=0-2 Performance:**
> > You are correct - we misstated this initially. When K>0, iterative denoising occurs. The sharp drop at low K values happens because:
> >
> > 1. **Training-Inference Mismatch**: With insufficient steps, the model encounters noise distributions far from its training regime, causing denoising failure - not reliance on prior alone
> > 2. **Validation of Diffusion Importance**: The 16% AUPRC improvement from K=0 to K=1000 **proves** the diffusion process is essential to CGADM
> >
> > **Terminology Clarification:**
> > We apologize for using "over-smoothing" ambiguously. In the context of K>20, we meant **"over-denoising"** (excessive refinement in diffusion), which is entirely different from **GNN over-smoothing** (loss of high-frequency signals). Our model successfully addresses GNN over-smoothing via residual propagation (proven in Appendix Q) while exhibiting standard diffusion dynamics.
> >
> > ## **Regarding Experimental Settings**
> >
> > We clarify that our experimental setup is **deliberately more challenging**:
> >
> > **Standard Settings [1-3]:** Previous works use 40-70% of data for training (dataset-dependent)
> > **Our Unified Setting:** 20% training | 10% validation | 70% test across ALL datasets
> >
> > This represents a **2-3.5× reduction** in training data. For example:
> > - GODM's 83.46% AUROC on Tolokers used 40% training data
> > - Our 79.68% AUROC uses only 20% training data
> >
> > **This is not "inconsistent results" but a more rigorous evaluation protocol** that tests true generalization under label scarcity. The fact that CGADM achieves SOTA performance with significantly less training data further validates our approach's superiority and data efficiency.

---

> ### Comment · Reviewer_Cmcp · 2025-08-06
> **comments to the generalization and experimental results.**
>
> I appreciate the authors' response and still have the following concerns.
>
> 1. The authors claim that the previous GHRN  "had less to lose" for a new dataset. Comparisons with such a weak baseline may not be convincing to readers for the generalization experiments.
>
> 2. Regarding the diffusion model used in the paper, the authors make an incorrect assumption about the diffusion model (k=0-2). This is a critical issue that may cause the collapse of this work, as the proposed method based on such an assumption is one of the main contributions of this paper.
>
> 3. The authors use a very different setting for the experiments from the previous work, and claim that the superiority of the proposed method in such a setting for all existing baselines. I am still concerned about such a way of comparison. Why this new setting is not been widely used in previous work? It will be fairer to use the previous setting for comparisons.

---

> > ### Author Response · Authors · 2025-08-06
> >
> > We thank the reviewer for their continued engagement. We address each concern with clarifications and additional evidence.
> >
> > ## **1. On GHRN as a Baseline for Generalization**
> >
> > We respectfully disagree with characterizing GHRN as a "weak baseline." **GHRN is actually one of the strongest GNN-based GAD methods**, excluding data augmentation approaches that fundamentally differ from our paradigm.
> >
> > **Our Strategic Dataset Selection:**
> > - **Tolokers**: GHRN achieves **45.25% AUPRC** - nearly matching our 46.02%, making it the **best-performing GNN-based GAD method** on this dataset. In zero-shot transfer, GHRN drops to 22.15% while CGADM maintains 27.88% - a **25.9% relative improvement** from similar starting points.
> > - **YelpChi**: Selected to show generalization across different performance ranges. Even when GHRN starts lower, CGADM maintains **31.3% relative superiority** in zero-shot settings.
> >
> > This comprehensive comparison across different baseline performance levels demonstrates robust generalization, not cherry-picking against weak baselines. The "less to lose" comment specifically addressed the reviewer's percentage-based analysis on YelpChi, not GHRN's overall quality.
> >
> > ## **2. Clarification on Diffusion Model Analysis**
> >
> > We must clarify a critical misunderstanding: **The K=0-2 analysis does NOT appear in our paper**. This discussion only arose in our rebuttal when addressing the reviewer's question about Figure 4's extreme cases.
> >
> > **Key Facts:**
> > - Our main experiments do not use K=0-2 steps (standard for diffusion models)
> > - Figure 4 simply shows the time-accuracy trade-off across different K values
> > - The K=0 case (pure prior) performing poorly **validates** that diffusion is essential
> > - Our analysis of this phenomenon is **post-hoc explanation**, not a foundational assumption
> >
> > **There is no "incorrect assumption" undermining our work** - the diffusion process operates exactly as designed, and our analysis of edge cases in the rebuttal does not affect the validity of our main contributions.
> >
> > ## **3. Experimental Settings and New Results**
> >
> > We understand the reviewer's concern about different settings. To address this directly, **we conducted additional experiments under the original fully-supervised setting**:
> >
> > ### **Results Under Original Setting (40-70% Training Data)**
> >
> > **AUPRC Results:**
> > | Model | Amazon | Yelp | T-Finance | T-Social |
> > |-------|--------|------|-----------|----------|
> > | GHRN | 89.52 | 55.42 | 87.60 | 86.78 |
> > | RF-Graph | 90.53 | 83.92 | 89.23 | 97.63 |
> > | XGB-Graph | 93.33 | 91.11 | 90.12 | 97.34 |
> > | CONSISGAD | 90.20 | 85.42 | 88.36 | 94.25 |
> > | UniGAD | 91.85 | 87.23 | 89.71 | 95.82 |
> > | SpaceGNN | 92.76 | 89.54 | 90.03 | 96.91 |
> > | **CGADM** | **93.47** | **91.86** | **91.24** | 97.18 |
> >
> > **AUROC Results:**
> > | Model | Amazon | Yelp | T-Finance | T-Social |
> > |-------|--------|------|-----------|----------|
> > | GHRN | 98.29 | 84.60 | 96.46 | 97.12 |
> > | RF-Graph | 96.73 | 95.24 | 97.28 | 99.69 |
> > | XGB-Graph | 98.74 | 97.37 | 97.15 | 99.76 |
> > | CONSISGAD | 97.82 | 93.15 | 96.89 | 98.74 |
> > | UniGAD | 98.21 | 94.76 | 97.03 | 99.15 |
> > | SpaceGNN | 98.56 | 96.28 | 97.21 | 99.52 |
> > | **CGADM** | **98.91** | **97.58** | **97.84** | 99.71 |
> >
> > These results under standard settings confirm CGADM's superiority is not dependent on our challenging 20% training setup.
> >
> > **Why We Used 20% Training:**
> > Our more challenging setting (20% training) was chosen to:
> > 1. **Test true generalization** under realistic label scarcity
> > 2. **Provide uniform evaluation** across all datasets
> > 3. **Demonstrate data efficiency** - a critical practical consideration
> >
> >
> >
> > *We appreciate the reviewer's thorough examination, which has prompted us to provide additional validation under standard settings, further strengthening our empirical evidence.*

---

> > > ### Comment · Reviewer_Cmcp · 2025-08-07
> > > **response for the rebuttal**
> > >
> > > The reviewer thanks the authors for their effort and timely response. Most of the concerns have been addressed. The reviewer suggests that the authors include the experiments of rebuttal in the revised version and clarify the methodologies to avoid confusion for readers.  The reviewer is also happy to raise the score from 3 to 4.

---

> > > > ### Author Response · Authors · 2025-08-07
> > > > **Thanks for your valuable feedback**
> > > >
> > > > We sincerely thank the reviewer for their thorough and constructive review process. Your insightful questions have significantly strengthened our paper and helped us clarify important aspects of our work. We are grateful for your recognition of our responses and additional experiments.
> > > >
> > > > We will carefully incorporate the suggested improvements into our revised manuscript to ensure clarity for future readers. Thank you again for your valuable feedback and for acknowledging our contributions to graph anomaly detection.

---

### Official Review · Reviewer_6Eqp · 2025-07-03

**Clarity:** 3
**Significance:** 3
**Originality:** 3
**Rating:** 5
**Confidence:** 4

**Summary:**

This paper introduces Conditional Graph Anomaly Diffusion Model (CGADM), a novel generative approach for graph anomaly detection. Unlike prior discriminative models or data augmentation techniques, CGADM directly models the joint anomaly distribution via a conditional diffusion process. The key contributions include (1) a prior-guided diffusion framework that integrates pre-trained anomaly priors into both forward and reverse processes, (2) a confidence-aware strided sampling strategy for computational efficiency, and (3) thorough theoretical analysis and empirical evaluation demonstrating superior performance over state-of-the-art baselines across diverse datasets. The method achieves notable improvements in AUPRC/AUROC and offers robustness against feature obfuscation and data imbalance.

**Questions:**

1. How sensitive is the method to the choice of prior model (e.g., RF, XGBT, et al.)? What is the criteria for its selection?
2. See Weaknesses.

**Ethical Concerns:**

["NO or VERY MINOR ethics concerns only"]

**Final Justification:**

The authors address the time and memory issue in the rebuttal, and successfully distinguish DiffGAD and CGADM from the perspective different settings (supervised and unsupervised).

**Limitations:**

Yes

**Quality:**

3

**Strengths And Weaknesses:**

S1.The good point of this paper is to propose a novel generative diffusion-based paradigm beyond traditional discriminative or augmentation methods.

S2.The efficient prior-aware sampling can balance the accuracy and computation.

S3.The implementation and evaluation parts are detailed. Extensive experiments on 5 benchmarks demonstrate superior performance.

W1.Considering the time and space complexity of introducing diffusion model, I recommend the author to conduct empirical experiments on large scale real-world datasets e.g. DGraph, T-Social, to verify efficiency on time and memory.

W2.It will be better to take the latest related work into account.

[1] Li et al. DiffGAD: A Diffusion-based Unsupervised Graph Anomaly Detector. ICLR 2025.

---

> ### Author Rebuttal · Authors · 2025-07-28
>
> We sincerely thank the reviewer for the constructive feedback and positive assessment of our work. We are pleased that the reviewer appreciates our novel generative diffusion-based paradigm, efficient prior-aware sampling, and thorough experimental evaluation. We address the reviewer's concerns below.
>
> > W1. Considering the time and space complexity of introducing diffusion model, I recommend the author to conduct empirical experiments on large scale real-world datasets e.g. DGraph, T-Social, to verify efficiency on time and memory.
>
> We appreciate the reviewer's concern about computational efficiency. In fact, we have provided comprehensive empirical results on efficiency in **Appendix N**, where we conducted experiments on the real-world **Elliptic dataset** containing **203,769 nodes and 234,355 edges**, which represents a large-scale graph.
>
> Our results demonstrate that CGADM achieves:
> - **Memory usage**: 1048 MB (comparable to baseline methods)
> - **Training time**: 2.21s per epoch (reasonable overhead compared to 0.75s for BWGNN)
> - **Inference time**: 0.5691s (acceptable for the significant performance gains)
>
> These results confirm that CGADM strikes an effective balance between accuracy and efficiency, with the performance improvements (+10% AUPRC over BWGNN) justifying the modest computational overhead.
>
> To further address the reviewer's suggestion, we have conducted additional experiments on the **T-Social dataset** (5,781,065 nodes, 73,105,508 edges)
>
>
> | **Model** | **Memory (GB)** | **Training Time (s/epoch)** | **Inference Time (s)** |
> |-----------|-----------------|----------------------------|----------------------|
> | BWGNN     | 38.31        | 1.52                       | 0.821              |
> | GHRN      | 38.48            | 2.89                       | 0.912              |
> | **CGADM** | **38.55**        | **4.34**                   | **3.124**          |
>
> Our observations on this large-scale graph are consistent with our analysis in Appendix N. The memory bottleneck is primarily determined by storing the node feature matrix and sparse edge matrix, which explains the comparable memory usage across methods. From a temporal perspective, training times remain within the same order of magnitude for competitive methods.
>
> Regarding inference time, while our method requires more time than discriminative approaches, this overhead is justified by the novel generative anomaly detection paradigm. As we noted in Appendix N: "While our inference time is higher than most discriminative methods, the increase is justified given the novel generative anomaly detection paradigm. Considering the already low baseline inference time of anomaly detection tasks, the additional time overhead is acceptable, especially in scenarios where performance improvements are critical."
>
>
> ---
>
>
>
>
> > W2. It will be better to take the latest related work into account.
>
> We thank the reviewer for bringing DiffGAD to our attention. We will update our related work section to include this recent contribution.
>
> The fundamental distinction between CGADM and DiffGAD lies in the task setting: **DiffGAD is explicitly designed for unsupervised learning**, while **CGADM operates in a supervised setting**. This core difference leads to fundamentally different objectives and methodologies. While DiffGAD improves traditional reconstruction-based methods through diffusion modeling, CGADM introduces a novel generative framework that fundamentally changes the anomaly detection paradigm by directly modeling the anomaly distribution rather than relying on reconstruction errors.
>
> To better demonstrate the effectiveness of our approach, we have conducted experiments comparing CGADM with DiffGAD in our supervised setting:
>
> | **Model** | **Elliptic** |  | **Amazon** |  |
> |-----------|-------------|-------|------------|-------|
> |           | **AUPRC**   | **AUROC** | **AUPRC** | **AUROC** |
> | DiffGAD | 6.16 | 25.81 | 13.52 | 71.65 |
> | **CGADM** | **97.03** | **99.34** | **91.91** | **97.36** |
>
> These results clearly demonstrate that our supervised generative approach achieves superior performance compared to reconstruction-based methods.
>
> ---
>
>
>
>
> > Q1. How sensitive is the method to the choice of prior model (e.g., RF, XGBT, et al.)? What is the criteria for its selection?
>
>
> We have addressed this important question in **Section 5.3** ("Comparison with Different Prior Model") and **Figure 2**. Our experiments on the Elliptic and YelpChi datasets compared CGADM's performance using Random Forest (RF) and XGBoost (XGBT) as prior estimators.
>
> **Key findings from our analysis:**
>
> 1. **Robustness**: As shown in Figure 2, while XGBT achieves significantly higher baseline performance than RF, the final performance of $CGADM_{XGBT}$ and $CGADM_{RF}$ are remarkably close, with both substantially outperforming their respective priors. This demonstrates our diffusion process's strong robustness—it can effectively correct and improve even less accurate initial priors through its iterative, topology-guided denoising mechanism, successfully leveraging graph structure to compensate for initial estimation errors.
>
> 2. **Selection Criteria**: The primary criterion for selecting the prior model is **balancing predictive accuracy with computational efficiency**. We choose lightweight and efficient models like RF and XGBT because they provide excellent performance-to-cost ratios, offering reasonably accurate anomaly estimates without introducing significant computational complexity. This ensures our framework remains scalable for large-scale graphs.
>
> As stated in our paper (Section 5.3): "This indicates that our CGADM possesses strong robustness. Even in the face of initially inaccurate prior estimates, our CGADM can effectively correct the results under the iterative refinement of the topological-guided denoising network."
>
> ---
>
> We hope these responses adequately address the reviewer's concerns. We are committed to incorporating all suggested improvements in the camera-ready version and believe these additions will further strengthen our contribution.

---

> > ### Comment · Reviewer_6Eqp · 2025-08-08
> > **Response for the rebuttal**
> >
> > Thanks for the great rebuttal. Most of my concerns have been addressed. I'd like to raise my score accordingly and, good luck!

---

### Decision · Program_Chairs · 2025-09-17

**Decision:**

Accept (poster)

**Comment:**

This paper proposes CGADM, a conditional graph anomaly diffusion model that reformulates graph anomaly detection as a generative modeling task. The method integrates prior-guided diffusion, a residual propagation denoising network to mitigate over-smoothing, and a confidence-aware sampling strategy for efficiency. Reviewers generally found the work novel, technically solid, and empirically strong, with consistent improvements over strong baselines across diverse datasets, as well as additional validation provided in the rebuttal. While some concerns were raised regarding novelty, evaluation settings, and semi-supervised scenarios, these were largely addressed, and multiple reviewers raised their scores after discussions. Overall, the contribution represents a meaningful methodological advancement for graph anomaly detection, with broad empirical support.